# Antagonistic control of DDK binding to licensed replication origins by Mcm2 and Rad53

**Syafiq Abd Wahab[1,2], Dirk Remus[1,2]\***

[1]Molecular Biology Program, Sloan Kettering Institute, Memorial Sloan Kettering Cancer Center, New York, United States; [2]Weill-Cornell Graduate School of Medical Sciences, New York, United States

**Abstract** Eukaryotic replication origins are licensed by the loading of the replicative DNA helicase, Mcm2-7, in inactive double hexameric form around DNA. Subsequent origin activation is under control of multiple protein kinases that either promote or inhibit origin activation, which is important for genome maintenance. Using the reconstituted budding yeast DNA replication system, we find that the flexible N-terminal extension (NTE) of Mcm2 promotes the stable recruitment of Dbf4-dependent kinase (DDK) to Mcm2-7 double hexamers, which in turn promotes DDK phosphorylation of Mcm4 and −6 and subsequent origin activation. Conversely, we demonstrate that the checkpoint kinase, Rad53, inhibits DDK binding to Mcm2-7 double hexamers. Unexpectedly, this function is not dependent on Rad53 kinase activity, suggesting steric inhibition of DDK by activated Rad53. These findings identify critical determinants of the origin activation reaction and uncover a novel mechanism for checkpoint-dependent origin inhibition.

## Introduction

To ensure the timely, accurate, and complete duplication of their genomes prior to cell division, eukaryotic cells initiate DNA replication at many replication origins distributed along the length of each chromosome. From each origin, two replication forks emanate in opposite direction to form a replication bubble. Although bidirectional origin firing has long been recognized as a universal feature of chromosomal DNA replication in all domains of life (*Huberman and Riggs, 1968*; *Prescott and Kuempel, 1972*), how pairs of oppositely oriented replication forks are established at chromosomal origins remains poorly understood. In eukaryotes, two copies of the replicative DNA helicase, Mcm2-7 (henceforth referred to as MCM), are loaded as a stable double-hexameric complex around double-stranded DNA (dsDNA) at the origin (*Evrin et al., 2009*; *Miller et al., 2019*; *Remus et al., 2009*). The MCM complex comprises six related proteins of the AAA+ family of ATPases that assemble into a hexameric ring with defined subunit order. Intriguingly, the two hexamers in a MCM double hexamer (DH) associate in a head-to-head configuration, thus providing a platform for the establishment of oppositely oriented sister replisomes. (*Abid Ali et al., 2017*; *Evrin et al., 2009*; *Li et al., 2015*; *Miller et al., 2019*; *Noguchi et al., 2017*; *Remus et al., 2009*). However, MCM DHs are catalytically inactive and require the regulated association of the essential helicase co-factors Cdc45 and GINS to form two active replicative DNA helicase complexes, termed CMG (Cdc45-MCM-GINS), which encircle single-stranded DNA (ssDNA) during unwinding (*Bell and Labib, 2016*). Therefore, to ensure bidirectional origin firing, mechanisms must exist that ensure simultaneous progression of both helicase complexes from the origin. As the head-to-head orientation of CMG helicases assembled at the origin requires them to pass each other during origin activation, it has been proposed that an active CMG encircling ssDNA is blocked by a dsDNA-encircling

**\*For correspondence:**
remusd@mskcc.org

**Competing interests:** The authors declare that no competing interests exist.

inactive CMG potentially formed around the opposite MCM hexamer, thereby imposing origin bidirectionality (*Douglas et al., 2018*; *Georgescu et al., 2017*).

The replication of chromosomes from multiple replication origins necessitates strict control mechanisms that prevent origins from re-firing within one cell cycle in order to maintain genome stability. Such re-replication control is achieved by a two-step mechanism that temporally separates helicase loading in late M and G1 phase from helicase activation in S phase (*Bell and Labib, 2016*). Helicase activation is controlled by two cell cycle-regulated protein kinases, Dbf4-dependent kinase (DDK) and cyclin-dependent kinase (CDK), which act in conjunction with a defined set of co-factors, comprising Sld3·7, Sld2, Dpb11, Pol ε, and Mcm10, in addition to GINS and Cdc45, to mediate CMG assembly (*Douglas et al., 2018*). The essential targets of CDK during CMG assembly are Sld2 and Sld3, which physically interact with distinct Dpb11 BRCT domains when phosphorylated to recruit GINS and Pol ε to the origin (*Bell and Labib, 2016*). The mechanism by which DDK promotes CMG assembly is somewhat obscure. Genetic and biochemical studies demonstrate that MCM subunits are the essential targets for DDK during DNA replication (*Deegan et al., 2016*; *Hardy et al., 1997*; *Randell et al., 2010*; *Sheu and Stillman, 2010*; *Yeeles et al., 2015*). Accordingly, DDK phosphorylation of the flexible N-terminal extensions (NTEs) of Mcm4 and −6 promotes the recruitment of Sld3·7 to MCM DHs by generating phosphorylation-dependent binding sites for Sld3, which in turn recruits Cdc45 (*Deegan et al., 2016*; *Gros et al., 2014*; *Heller et al., 2011*; *Tanaka et al., 2011*; *Yeeles et al., 2015*). However, the nature and stoichiometry of the Sld3 interaction with MCM DHs is unclear, as Sld3 exhibits limited sequence specificity for phospho-dependent binding sites, and phospho-mimicking mutations in either Mcm4 or −6 are sufficient to bypass the requirement for DDK (*Deegan et al., 2016*; *Randell et al., 2010*). Moreover, other mutations in Mcm5 and Mcm4 that do not involve phospho-mimetic amino acid substitutions can also bypass the requirement for DDK (*Hardy et al., 1997*; *Sheu and Stillman, 2010*). It is not known, if these mutations allow Sld3 recruitment in the absence of MCM phosphorylation, or if they bypass the requirement for Sld3 altogether.

Replication origins do not fire simultaneously upon S phase entry, but in a staggered programmatic manner throughout S phase (*Rhind and Gilbert, 2013*). The replication timing program is in part imposed by limiting concentrations of the initiation factors Sld2, Dpb11, Sld3, Cdc45, and Dbf4, which restrict CMG assembly to subsets of origins throughout S phase (*Mantiero et al., 2011*; *Tanaka et al., 2011*). While limiting initiation factors that interact transiently with origins are thought to be recycled from early to late origins in a normal S phase, such recycling is prevented by the S phase checkpoint to inhibit late origin firing. The S phase checkpoint is a vital kinase signaling cascade that induces a range of cellular responses in addition to the inhibition of origin firing, including increasing dNTP levels, stabilization of stalled replication forks, and DNA repair to promote genome maintenance during replication stress (*Pardo et al., 2017*).

In budding yeast, the minimal targets for checkpoint-dependent origin inhibition are Sld3 and Dbf4, which are substrates for the checkpoint effector kinase, Rad53 (*Lopez-Mosqueda et al., 2010*; *Zegerman and Diffley, 2010*). Rad53 phosphorylation of Sld3 inhibits its physical interactions with MCM, Cdc45, and Dpb11 (*Deegan et al., 2016*; *Lopez-Mosqueda et al., 2010*; *Zegerman and Diffley, 2010*). How Rad53 inhibits Dbf4 is not clear. Rad53 was found to inhibit DDK kinase activity *in vitro*, but the mechanism of inhibition is unknown (*Kihara et al., 2000*; *Weinreich and Stillman, 1999*). Moreover, DDK activity in these studies was assessed using isolated Mcm2 or Mcm7 subunits as substrates or by measuring DDK autophosphorylation (*Kihara et al., 2000*; *Weinreich and Stillman, 1999*). However, since DDK exhibits high specificity for Mcm4 and −6 in the context of MCM DHs, the effect of Rad53 on DDK activity at licensed origins remains to be determined (*Francis et al., 2009*; *Randell et al., 2010*; *Sheu and Stillman, 2006*; *Sun et al., 2014*). Rad53 has also been reported to disrupt DDK-chromatin association in cells treated with hydroxyurea (HU) (*Pasero et al., 1999*). DDK binds to chromatin via an interaction with MCM at licensed replication origins (*Dowell et al., 1994*; *Francis et al., 2009*; *Jares and Blow, 2000*; *Jares et al., 2004*; *Sato et al., 2003*; *Sheu and Stillman, 2006*; *Takahashi and Walter, 2005*; *Weinreich and Stillman, 1999*; *Yanow et al., 2003*). Several potential DDK docking sites have been identified in MCM based on pairwise interaction studies with individual MCM subunits (*Ramer et al., 2013*; *Sheu and Stillman, 2006*). However, it has not been tested to what extent these contribute to DDK binding in the context of the MCM DH, which is known to greatly stimulate DDK substrate specificity and kinase

efficacy (*Francis et al., 2009*; *Sun et al., 2014*). How Rad53 may affect the DDK-chromatin association has not been addressed.

Here, we employ the reconstituted budding yeast DNA replication system to investigate the mechanism of DDK docking to MCM DHs and its regulation by Rad53 (*Devbhandari et al., 2017*; *Devbhandari and Remus, 2020*). We find that the flexible Mcm2 NTE is necessary for DDK docking onto MCM DHs, Mcm4 and −6 phosphorylation, and origin firing. The data suggests that the NTEs of both Mcm2 protomers in the MCM DH are required for efficient origin activation. Intriguingly, activated Rad53 can form a physical complex with DDK and disrupt DDK binding to MCM DHs in a manner that is independent of Rad53 kinase activity. These observations have important implications for the mechanism of bidirectional origin firing and its regulation by the replication checkpoint.

## Results

### Residues 1–127 of Mcm2 are dispensable for MCM DH assembly and stability

While the structured N-terminal domains (NTDs) and AAA+ ATPase domains are conserved between archaeal and eukaryotic MCM proteins, the long unstructured NTEs of Mcm2, −4, and −6 are unique to eukaryotes. The NTEs of Mcm4 and −6 play a fundamental role during DNA replication by serving as phospho-acceptors for DDK kinase during origin activation (*Deegan et al., 2016*; *Francis et al., 2009*; *Randell et al., 2010*; *Sheu and Stillman, 2006*). In contrast, the Mcm2 NTE contains a nuclear localization sequence (NLS) that mediates nuclear import of Cdt1·MCM (*Liku et al., 2005*) and a conserved histone H3/H4 binding domain (HBD) that controls nucleosome segregation at replication forks (*Foltman et al., 2013*; *Gan et al., 2018*; *Huang et al., 2015*; *Petryk et al., 2018*). As nuclear import is irrelevant for DNA replication with purified proteins, and chromatin can be omitted from DNA replication reactions *in vitro*, we asked whether the Mcm2 NTE possesses a basic DNA replication function. For this, we engineered a TEV protease cleavage site downstream of the HBD, between residues A127 and Y128 (Mcm2-TEV; *Figure 1A*). Purified Cdt1·Mcm2-7 complexes harboring Mcm2-TEV (Cdt1·MCM$^{2\text{-TEV}}$) are quantitatively and specifically cleaved at the engineered TEV site by TEV protease, resulting in Cdt1·MCM complexes containing the Mcm2-Δ127 N-terminal truncation, whereas wildtype Cdt1·MCM complexes are resistant to TEV protease cleavage (*Figure 1B*, *Figure 1—figure supplement 1*).

First, we tested if proteolytic truncation of the Mcm2 NTE affects MCM DH stability. For this, we performed reconstituted MCM loading reactions on origin-containing DNA using purified ORC, Cdc6, and either wildtype Cdt1·MCM or Cdt1·MCM$^{2\text{-TEV}}$ (*Remus et al., 2009*). Following MCM loading, DNA-bound complexes were washed to remove free proteins, treated with TEV protease, and analyzed by SDS-PAGE. The DNA was immobilized on paramagnetic streptavidin-coated beads via a single photo-cleavable 5'-terminal biotin moiety to allow elution of the DNA from the beads with UV light for analysis. To differentiate MCM DHs loaded around DNA from potential other more loosely associated complexes, such as the OCCM (O̲RC-C̲dc6-C̲dt1-M̲CM) (*Yuan et al., 2017*), DNA-bound complexes were washed with a high-salt buffer prior to DNA elution (*Remus et al., 2009*). While MCM DHs are resistant to salt-elution from the DNA, loading factors and loosely associated MCM complexes are efficiently disrupted by stringent salt washes. In addition, control MCM loading reactions were carried out in the presence of non-hydrolyzable ATPγS as MCM DH formation is strictly dependent on ATP hydrolysis (*Bell and Labib, 2016*; *Remus et al., 2009*). Using this approach, we demonstrate comparable DNA loading efficiencies for wildtype Cdt1·MCM and Cdt1·MCM$^{2\text{-TEV}}$ (*Figure 1C*). Moreover, truncation of the Mcm2 NTE from Mcm2-TEV-containing DHs, despite being efficient, did not negatively affect MCM DH retention on DNA. Thus, maintenance of MCM DHs is not dependent on residues 1–127 of the Mcm2 NTE.

Next, we tested if Mcm2 residues 1–127 are important for MCM loading during pre-RC formation. For this we digested Cdt1·MCM$^{2\text{-TEV}}$ with TEV protease and re-purified the resulting Cdt1·MCM$^{2\Delta127}$ complex from the digestion reaction by gel-filtration chromatography. Truncation of the Mcm2 N-terminus did not have a noticeable effect on Cdt1·MCM stability, as Cdt1·MCM$^{2\Delta127}$ eluted in a mono-disperse peak at the expected position for the Cdt1·MCM heptamer during gel-filtration (*Figure 1D*). Importantly, Cdt1·MCM$^{2\Delta127}$ did not exhibit a discernible MCM loading defect,

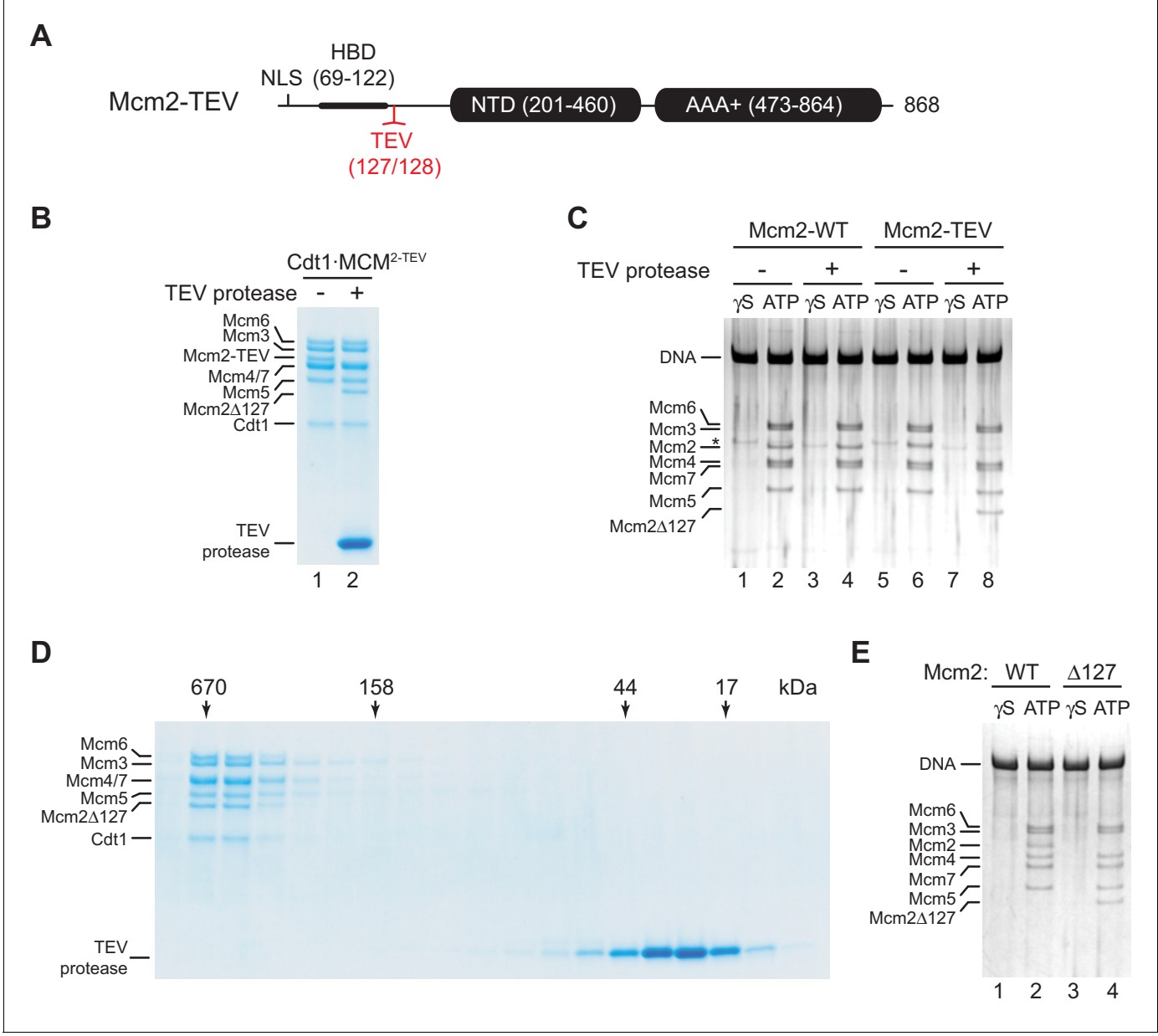

**Figure 1.** Residues 1–127 of the Mcm2 NTE are dispensable for MCM DH stability. (**A**) Schematic of Mcm2 domain structure. Numbers indicate amino acid positions. The position of the TEV cleavage site is highlighted in red. NLS: Nuclear localization sequence; HBD: Histone binding domain; NTD: N-terminal domain; AAA+: ATPase domain. (**B**) Cdt1·MCM²⁻ᵀᴱⱽ was mock-treated or digested with TEV protease for 1 hr at 30°C, as indicated. Reactions were fractionated on SDS-PAGE and stained with Coomassie blue. (**C**) MCM loading reactions were performed on 3 kbp ARS305-containing DNA in the presence of ATPγS (γS) or ATP as indicated. DNA-bound material was washed with high-salt buffer, mock-treated or digested with TEV protease as indicated, washed again with high-salt buffer, and analyzed by SDS-PAGE and silver staining. * denotes Orc1 protein. (**D**) Gel-filtration analysis of purified Cdt1·MCM²⁻ᵀᴱⱽ following digestion with TEV protease. The digestion reaction was fractionated on a Superdex 200 column and fractions analyzed by SDS-PAGE and Coomassie stain. (**E**) Mcm2-7 loading reactions with either wildtype Cdt1·MCM (lanes 1+2) or Cdt1·MCM²⁻Δ¹²⁷ (lanes 3+4). Reactions were performed either in the presence of ATPγS or ATP as indicated and DNA-beads subsequently washed with high-salt buffer. DNA-bound fractions were analyzed by SDS-PAGE and silver stain.

The online version of this article includes the following source data and figure supplement(s) for figure 1:

**Source data 1.** *Figure 1 B+D*.
**Source data 2.** *Figure 1C*.
**Source data 3.** *Figure 1E*.
**Figure supplement 1.** Time course analysis of Cdt1·MCM²⁻ᵀᴱⱽ and Cdt1·MCM²⁻ᵂᵀ cleavage by TEV protease.
**Figure supplement 1—source data 1.** *Figure 1—figure supplement 1*.

demonstrating that the Mcm2 N-terminus is dispensable for MCM recruitment and MCM DH assembly around DNA (*Figure 1E*).

## Residues 1–127 of Mcm2 are important for DNA replication

Next we tested if residues 1–127 of the Mcm2 NTE are required for DNA replication *in vitro*. For this, we performed DNA replication reactions both on naked DNA templates and reconstituted chromatin as described previously (*Devbhandari et al., 2017*; *Devbhandari and Remus, 2020*). As FACT and Nhp6 have been demonstrated to promote replisome progression through chromatin *in vitro* (*Kurat et al., 2017*), purified FACT and Nhp6 were also included in chromatin replication

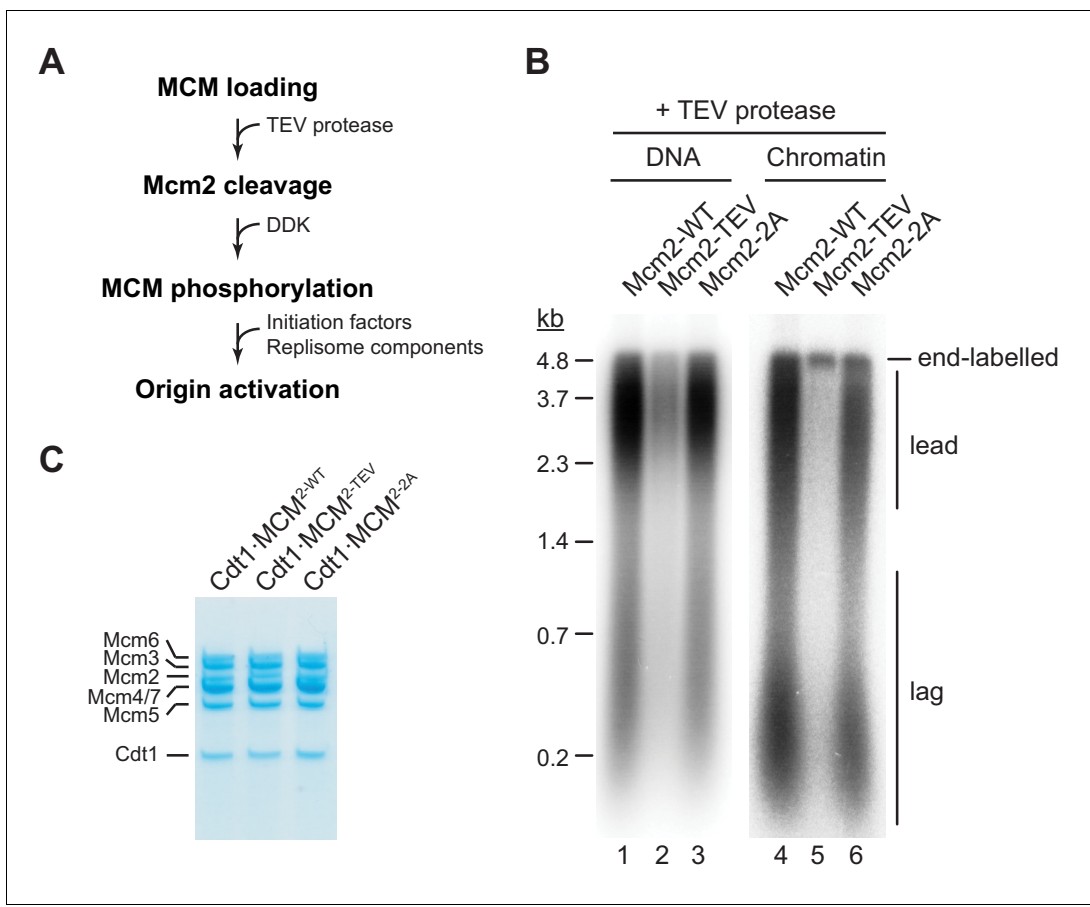

**Figure 2.** The Mcm2 NTE is important for DNA replication. (**A**) Experimental outline. (**B**) *In vitro* DNA replication reactions were performed on naked (lanes 1–3) or chromatinized (lanes 4–6) circular plasmid DNA (p1017, 4.8 kbp). TEV protease was added to each reaction following MCM loading for 1 hr at 30°C, before addition of DDK and standard initiation/replisome factors. Chromatin replication reactions additionally contained FACT and Nhp6. Products were analyzed by 0.8% denaturing agarose gel-electrophoresis and autoradiography. Lead: Leading strand product; lag: Lagging strand product. (**C**) Purified Cdt1·MCM complexes containing either wildtype Mcm2 (Cdt1·MCM$^{2\text{-WT}}$), Mcm2-TEV (Cdt1·MCM$^{2\text{-TEV}}$), or Mcm2-2A (Cdt1·MCM$^{2\text{-2A}}$).

The online version of this article includes the following source data and figure supplement(s) for figure 2:

**Source data 1.** *Figure 2A*.
**Source data 2.** *Figure 2C*.
**Figure supplement 1.** FACT/Nhp6-dependent chromatin replication.
**Figure supplement 1—source data 1.** *Figure 2—figure supplement 1A*.
**Figure supplement 1—source data 2.** *Figure 2—figure supplement 1B*.
**Figure supplement 2.** Attenuation of DNA synthesis in the presence of Mcm2-TEV is dependent on TEV protease cleavage.
**Figure supplement 2—source data 1.** *Figure 2—figure supplement 2*.

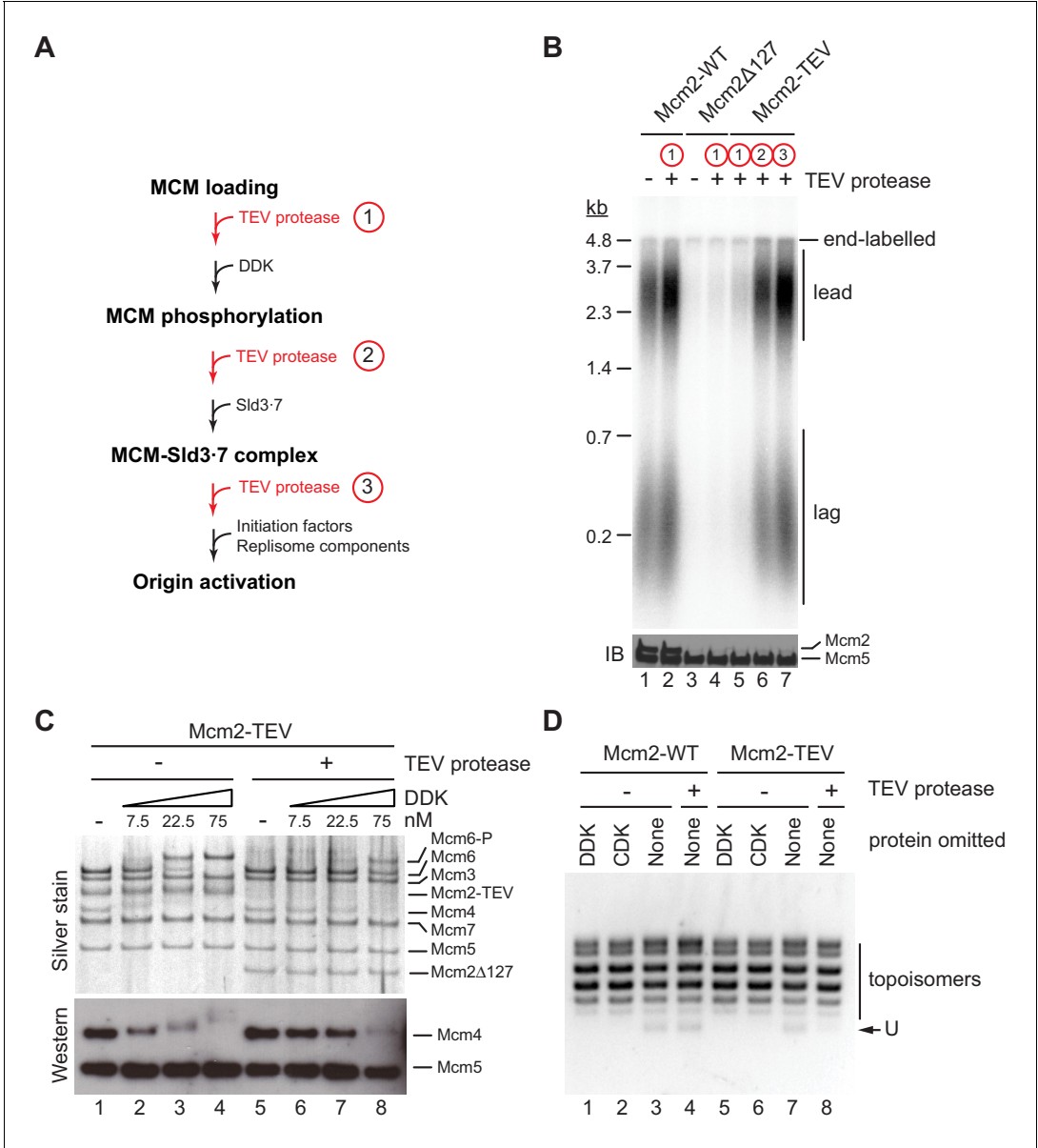

**Figure 3.** The Mcm2 NTE promotes DDK function during origin activation. (A) Experimental outline for experiment in B. Variable addition points for TEV protease are highlighted in red. (B) Standard *in vitro* DNA replication reactions were performed using p1017 (4.8 kb) as a template. TEV protease or mock buffer was added for 1 hr at 30°C as indicated. Reaction products were analyzed by denaturing agarose gel-electrophoresis and autoradiography (top). A fraction of each reaction was analyzed by SDS-PAGE and western blot using antibodies against Mcm2 and Mcm5 (bottom); note that the N-terminal epitope recognized by the Mcm2 antibody is lost after TEV protease cleavage. (C) MCM DHs assembled with Cdt1·MCM[2-TEV] were either mock-treated (lanes 1–4) or digested with TEV protease (lanes 5–8). DDK was subsequently added to the reactions at the indicated concentrations and reactions analyzed by SDS-PAGE and silver stain or western blot using antibodies against Mcm4 and Mcm5. (D) Plasmid unwinding assay. CMGs were assembled with Cdt1·MCM[2-WT] (lanes 1–4) or Cdt1·MCM[2-TEV] (lanes 5–8) using p79 (3 kbp) as substrate. TEV protease was added to the reactions after the MCM loading step, prior to the addition of DDK, CDK, Sld2, Sld3·7, Dpb11, GINS, Cdc45, Pol ε, RPA, and Mcm10 as indicated. DNA was repurified from the reaction and analyzed by native agarose gel-electrophoresis and EtBr stain. U: U-form DNA.

The online version of this article includes the following source data for figure 3:

**Source data 1.** *Figure 3B*, autoradiograph.
**Source data 2.** *Figure 3B*, immunoblot: Mcm2, Mcm5.
**Source data 3.** *Figure 3C*, silver stain.
**Source data 4.** *Figure 3C*, immunoblot: Mcm4, Mcm5.
**Source data 5.** *Figure 3D*.

reactions here (*Figure 2—figure supplement 1*). TEV protease cleavage of the Mcm2 NTE was induced following MCM loading (*Figure 2A*).

Intriguingly, truncation of Mcm2 residues 1–127 largely attenuated DNA replication (*Figure 2B*). Inhibition of DNA replication in the presence of Mcm2-TEV was dependent on TEV protease (*Figure 2—figure supplement 2*). The DNA replication defect was not due to a loss of HBD function, as mutation of two conserved tyrosine residues, Y82 and Y91, in the Mcm2 HBD (Cdt1·MCM$^{2-2A}$, *Figure 2C*) that have been previously shown to disrupt histone H3/H4 and FACT binding to Mcm2 has little effect on DNA replication using either DNA or chromatin as a template (*Foltman et al., 2013*). This is consistent with previous reports demonstrating that yeast cells harboring the *mcm2-2A* allele are viable and exhibit only mild chromosome replication defects (*Foltman et al., 2013*). This data reveals that the Mcm2 NTE performs a fundamental function during normal DNA replication that is distinct from its histone H3/H4 chaperone activity.

## The Mcm2 NTE promotes DDK function during the initiation of DNA replication

In order to dissect which step in the DNA replication reaction is defective in the absence of residues 1–127 of Mcm2, we performed order-of-addition experiments by adding TEV protease at various steps of the origin firing pathway (*Figure 3A*). In control experiments, TEV protease did not disrupt the DNA replication proficiency of wild-type Cdt1·MCM when added prior to DDK immediately after MCM loading (*Figure 3B*). Conversely, purified Cdt1·MCM$^{2Δ127}$ was deficient for DNA replication irrespective of the addition of TEV protease, as expected. As before, in the presence of Cdt1·MCM$^{2-TEV}$, addition of TEV protease to the reaction immediately after MCM loading inhibited origin activation. In striking contrast, addition of TEV protease after DDK or Sld3·7 allowed DNA replication to proceed normally. This data indicates that Mcm2 residues 1–127 promote DDK function during the initiation of DNA replication but are dispensable for DNA replication after the Sld3 step of the initiation reaction.

As the essential function of DDK is the phosphorylation of the NTEs of Mcm4 and Mcm6, we asked whether proteolytic truncation of the Mcm2 NTE affects Mcm4 and −6 phosphorylation by DDK. For this, we assembled MCM DHs from Cdt1·MCM$^{2-TEV}$ on bead-immobilized DNA and monitored DDK phosphorylation-dependent gel-mobility shifts of MCM subunits by SDS-PAGE (*Figure 3C*). MCM DHs harboring Mcm2-TEV were either digested or mock-treated with TEV protease prior to DDK addition. In the presence of the full-length Mcm2 NTE, Mcm4 and Mcm6, but not any of the other MCM subunits, exhibited a pronounced retardation in gel-mobility in the presence of DDK. The disappearance of the unphosphorylated Mcm4 and −6 bands at higher DDK concentrations demonstrates that these subunits were phosphorylated quantitatively by DDK. In contrast, the DDK-dependent gel-retardation of Mcm4 and −6 was strongly diminished at all DDK concentrations tested when the Mcm2 NTE was truncated with TEV protease prior to DDK addition. Thus, the Mcm2 NTE promotes the phosphorylation of Mcm4 and Mcm6 by DDK in the context of MCM DHs.

Phosphorylation of the Mcm4 and −6 NTEs by DDK promotes the assembly of the CMG helicase. We, therefore, tested if truncation of the Mcm2 NTE impairs CMG assembly using an origin-dependent CMG helicase assay that detects CMG helicase activity by the generation of highly unwound circular plasmid DNA, termed U-form DNA (*Douglas et al., 2018*). As expected, generation of U-form DNA in the presence of either Mcm2-WT or Mcm2-TEV is dependent on both CDK and DDK, demonstrating that plasmid unwinding is dependent on CMG assembly (*Figure 3D*). Importantly, generation of U-form DNA was suppressed specifically in the presence of Mcm2-TEV when TEV protease was added to the reaction after the MCM loading step, prior to the addition of DDK and other initiation factors. This data is consistent with residues 1–127 of the Mcm2 NTE promoting CMG assembly. In summary, we conclude that the Mcm2 NTE is important for DNA replication by promoting the phosphorylation of Mcm4 and −6 by DDK and subsequent CMG assembly, whereas it is dispensable for DNA replication after CMG assembly.

## The Mcm2 NTE promotes DDK docking onto MCM DHs

Next, we addressed how Mcm2 may promote the phosphorylation of Mcm4 and −6 by DDK. Previous studies had proposed a docking mechanism by which a stable association of DDK with MCM DHs promotes processive multi-site phosphorylation of the Mcm4 and −6 NTEs (*Francis et al.,*

*2009*; *Sheu and Stillman, 2006*). These studies, however, either utilized complex cell extracts to load MCM onto DNA, which may include unknown proteins that bridge the DDK-MCM interaction, or examined the binding of DDK to the isolated Mcm4 subunit, leaving open the question how DDK interacts with MCM subunits in the context of the MCM DH. We, therefore, investigated the direct binding of DDK to purified MCM DHs *in vitro*. For this, MCM DHs bound to DNA immobilized on paramagnetic beads were isolated from MCM loading reactions, washed with high-salt buffer, and incubated under various conditions with purified DDK.

Titration of DDK into a MCM DH binding reaction revealed that DDK binding to the DNA beads started to saturate at 75 nM DDK (*Figure 4A*). Importantly, when MCM DH formation was prevented by omission of Cdt1·MCM from the MCM loading reaction DDK binding to the DNA beads was not observed even at 300 nM DDK, demonstrating that DDK associates with MCM DHs on the DNA. To determine if DDK binds specifically to MCM DHs, we performed MCM loading reactions in the presence of ATP or ATPγS in order to assemble MCM DHs or OCCMs, respectively (*Figure 4B*). Unlike the MCM DH, the OCCM contains only a single MCM hexamer partially loaded around DNA (*Yuan et al., 2017*). Because the OCCM is sensitive to high-salt buffer washes, free proteins were removed from the reactions by low-salt buffer washes and DNA-bound complexes were subsequently incubated with DDK in the presence of ATPγS. DDK binding was exclusively observed in the presence of MCM DHs; neither OCCM nor the individual loading factors ORC, Cdc6, or Cdt1·MCM, or pairwise combinations of these, formed stable complexes with DDK on DNA. Thus, DDK-MCM complex formation is strictly dependent on MCM DH assembly.

Next, we tested the contribution of the Mcm2 NTE to the DDK-MCM DH interaction. To this end, MCM DHs harboring Mcm2-TEV were incubated with TEV protease prior to addition of DDK. As before, proteolytic removal of residues 1–127 from the Mcm2 N-terminus did not affect MCM DH stability on the DNA (*Figure 4C*). However, truncation of the Mcm2 N-terminus resulted in severely diminished binding of DDK to MCM DHs, demonstrating that Mcm2 residues 1–127 are important for the association of DDK with MCM DHs. To determine if the Mcm2 N-terminus is important only for the recruitment of DDK or also for the retention of DDK on MCM DHs, TEV protease was added to the reaction either before or after DDK binding. As shown in *Figure 4D*, while truncation of the Mcm2 NTE prior to the addition of DDK inhibited DDK-MCM DH complex formation as before, addition of TEV protease after DDK-MCM DH complex formation also led to the release of DDK from MCM DHs, demonstrating that the Mcm2 NTE is required for the retention of DDK on MCM DHs.

The DDK-MCM DH complex was remarkably resistant to extensive buffer washes including up to 0.3 M KOAc, attesting to the relative stability of the DDK-MCM DH interaction (*Figure 4E*). Disruption of the complex was observed in wash buffer containing 0.5 M NaCl, a condition in which MCM DHs remain stably bound to DNA, indicating that electrostatic interactions play an important role in mediating the DDK-MCM DH interaction. In the above experiments DDK binding to MCM DHs was monitored in the presence of the non-hydrolyzable ATP analogue ATPγS with the intention to trap DDK bound to MCM DH. Indeed, reduced DDK binding to MCM DHs was observed in the absence of nucleotide relative to the level of DDK binding to MCM DHs in the presence of ATP, ATPγS, or the non-hydrolyzable ATP analogue adenylyl-imidodiphosphate (AMP-PNP), indicating that ATP binding promotes DDK-MCM DH complex formation (*Figure 4F*). In the presence of ATP, both Cdc7 and Dbf4 exhibit a pronounced retardation in gel-mobility due to the auto-phosphorylation activity of DDK (*Hughes et al., 2010*; *Weinreich and Stillman, 1999*). Intriguingly, the level of DDK binding to MCM DHs was essentially identical in the presence of ATP, ATPγS, or AMP-PNP. This demonstrates that DDK docking onto MCM DHs is independent of Mcm4 and −6 phosphorylation. Moreover, these observations reveal that DDK remains stably bound to the MCM DHs after phosphorylation of the Mcm4 /-6 NTEs. Together, these results demonstrate that the Mcm2 NTE is required for the physical interaction of DDK with MCM DHs, explaining the inhibition of Mcm4 and −6 phosphorylation in its absence.

## Efficient origin activation requires the Mcm2 N-termini of both MCM hexamers

Bidirectional origin firing requires the activation of both MCM hexamers in a MCM DH to form a pair of oppositely oriented replication forks. We, therefore, wanted to determine if both Mcm2 N-termini of a MCM DH are required for origin activation. To this end, we assembled mixed MCM DHs from

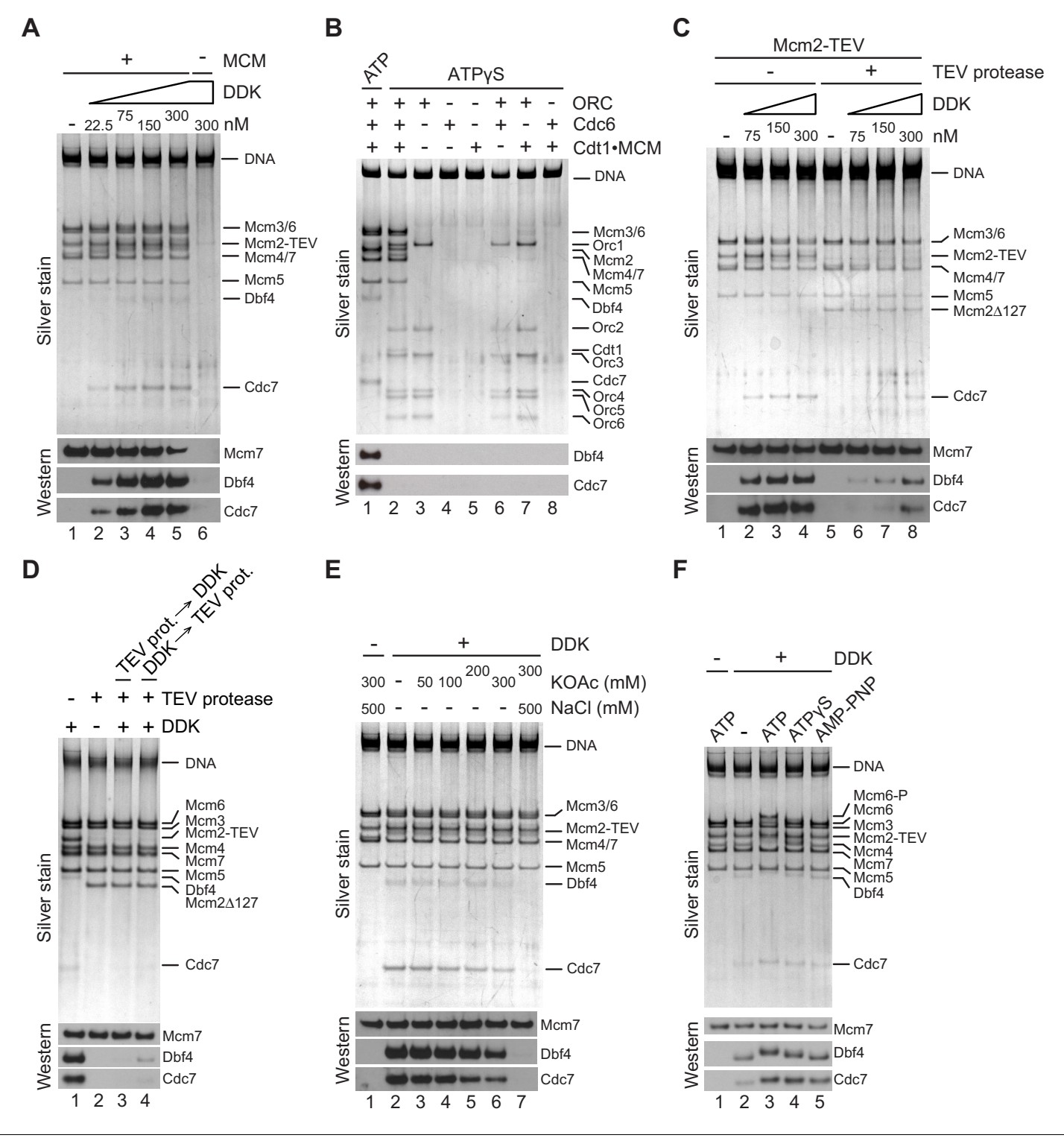

**Figure 4.** The Mcm2 NTE promotes binding of DDK to MCM DHs. (**A**) MCM DHs were assembled on bead-immobilized DNA, washed with high-salt buffer, and subsequently incubated with ATPγS and DDK at the indicated concentrations. As a control, Cdt1·MCM was omitted from the MCM loading reaction in lane 6. After incubation with DDK, DNA-bound material was isolated and analyzed by SDS-PAGE and silver stain (top) or western blot using antibodies against Mcm7, Dbf4, or Cdc7 (bottom). (**B**) MCM loading reactions were carried out either in the presence of ATP (lane 1) or ATPγS (lanes 2–8). DNA beads were subsequently washed with low-salt buffer and incubated with DDK in the presence of ATPγS. DNA-bound material was analyzed as in A. (**C**) MCM DHs were assembled from Cdt1·MCM$^{2\text{-TEV}}$, mock-treated or digested with TEV protease as indicated and incubated with purified DDK

*Figure 4 continued on next page*

*Figure 4 continued*

at the indicated concentrations. DNA-bound material was analyzed as in A. (**D**) MCM DHs were assembled from Cdt1·MCM$^{2\text{-TEV}}$ and mock-treated or digested with TEV protease as indicated. In lane 3, DDK was added after TEV protease, in lane 4 DDK was added before TEV protease. DDK was included at 150 nM. DNA-bound material was analyzed as in A. (**E**) DNA-bound DDK-MCM DH complexes were washed with buffer containing the indicated concentration of KOAc, and where indicated followed by a wash with buffer containing 500 mM NaCl. (**F**) MCM DHs were assembled on bead-immobilized DNA, washed to remove free ATP, and subsequently incubated with DDK in the presence of ATP or ATP analogues, as indicated. DNA-bound material was analyzed as in A.

The online version of this article includes the following source data for figure 4:

**Source data 1.** *Figure 4A*, silver stain.
**Source data 2.** *Figure 4 A+E*, immunoblot: Mcm7, Cdc7.
**Source data 3.** *Figure 4 A+E*, immunoblot: Dbf4.
**Source data 4.** *Figure 4B*, silver stain.
**Source data 5.** *Figure 4B*, immunoblot: Cdc7.
**Source data 6.** *Figure 4B*, immunoblot: Dbf4.
**Source data 7.** *Figure 4C*, silver stain.
**Source data 8.** *Figure 4C*, immunoblot: Mcm7, Cdc7.
**Source data 9.** *Figure 4C*, immunoblot: Dbf4.
**Source data 10.** *Figure 4D*, silver stain.
**Source data 11.** *Figure 4D*, immunoblot: Mcm7.
**Source data 12.** *Figure 4D*, immunoblot: Dbf4.
**Source data 13.** *Figure 4D*, immunoblot: Cdc7.
**Source data 14.** *Figure 4E*, silver stain.
**Source data 15.** *Figure 4F*, silver stain.
**Source data 16.** *Figure 4F*, immunoblot: Cdc7.
**Source data 17.** *Figure 4F*, immunoblot: Mcm7.
**Source data 18.** *Figure 4F*, immunoblot: Dbf4.

both wildtype Cdt1·MCM and Cdt1·MCM$^{2\Delta127}$. A 1:1 mixture of wildtype Cdt1·MCM and Cdt1·MCM$^{2\Delta127}$ yields a mixed DH population composed of 25 % WT/WT, 50 % WT/Δ127, and 25 % Δ127/ Δ127 with respect to Mcm2. Accordingly, if a single Mcm2 NTE per MCM DH is sufficient for maximal origin activity, a 25 % reduction in origin activity would be expected for a 1:1 mixture of wildtype Cdt1·MCM and Cdt1·MCM$^{2\Delta127}$. Conversely, a 75 % loss in origin activity would be expected if both Mcm2 NTEs of a MCM DH were required for origin firing.

We performed standard DNA replication reactions including wildtype or mutant Cdt1·MCM complexes at 80 nM during the MCM loading step, a concentration that supports near maximal DNA synthesis. Importantly, at Cdt1·MCM concentrations below 80 nM origin firing efficiency strongly correlates with Cdt1·MCM concentration, thus allowing sensitive detection of loss of MCM DH activity (**Figure 5—figure supplement 1A**). Similar to our previous approach (**Figure 2**), total DNA synthesis in the complete absence of the Mcm2 residues 1–127 was reduced by ~75% relative to that in the presence of full-length Mcm2 (**Figure 5 A+B**). This DNA synthesis defect was due to a defect in origin activation and not fork progression as leading and lagging strand lengths were similar in both conditions (**Figure 5C**). Importantly, DNA replication was reduced by ~50% at a 1:1 ratio of Cdt1·MCM$^{2\Delta127}$ to wildtype Cdt1·MCM, well beyond the 25% reduction expected if one Mcm2 NTE was sufficient for normal origin activation. The fact that the loss in DNA replication levels falls short of the 75% reduction expected if both Mcm2 NTEs in the Mcm2-7 DH were required for origin firing is attributable to the residual origin activity observed in the absence of Mcm2 NTEs (**Figure 5—figure supplement 1B**).

Importantly, we did not detect any evidence for asymmetric, or unidirectional, origin firing arising from the activation of a single MCM hexamer within heterologous MCM DHs assembled from both Cdt1·MCM$^{2\text{-WT}}$ and Cdt1·MCM$^{2\Delta127}$. The normal half-unit length leading strand length obtained on circular DNA templates results from the termination of DNA replication at the plasmid pole opposite the replication origin when sister replication forks emanating from a replication origin converge. Consequently, in the absence of an opposing replication forks, a single fork emanating from an origin would be able to traverse past the half-unit length of a circular plasmid, giving rise to > half unit length leading strand lengths. The lane scan in **Figure 5C** demonstrates that > half unit length

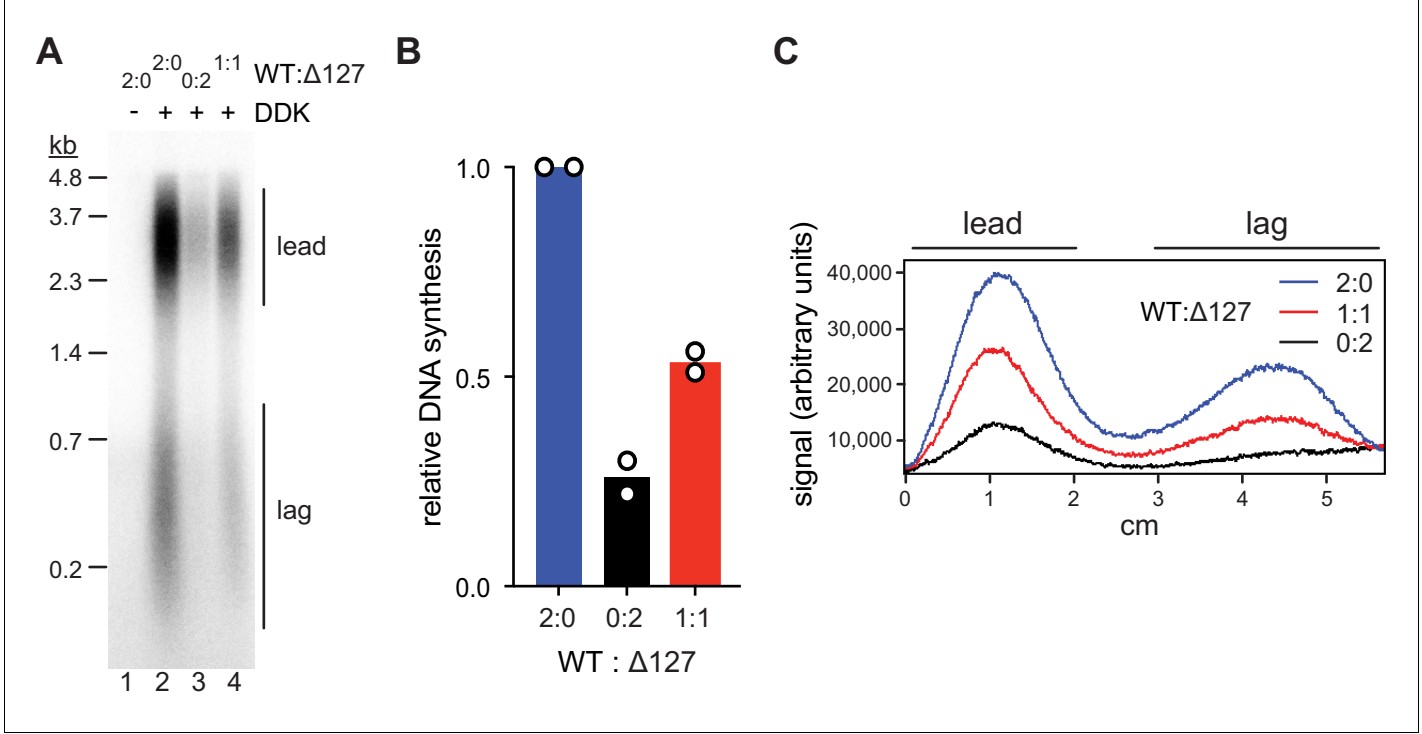

**Figure 5.** Mcm2-WT does not rescue the Mcm2Δ127 replication defect. (**A**) Standard DNA replication reaction using p1017 (4.8 kb) as template. Cdt1·MCM$^{2-Δ127}$ and Cdt1·MCM$^{2-WT}$ were included at the MCM loading step at the indicated ratios; the total concentration of Cdt1·MCM was 80 nM in the Mcm2-7 loading reaction. (**B**) Quantification of total relative DNA synthesis in reactions of experiment in C. Bars represent the average of two independent experiments. (**C**) Lane traces of experiment in C.

The online version of this article includes the following source data and figure supplement(s) for figure 5:

Source data 1. *Figure 5A*, autoradiograph.
Figure supplement 1. The effect of Cdt1·MCM and DDK concentrations on DNA replication *in vitro*.
Figure supplement 1—source data 1. *Figure 5—figure supplement 1A*.
Figure supplement 1—source data 2. *Figure 5—figure supplement 1A*.

leading strands are not synthesized in the presence of mixed MCM DHs. Alternatively, as the leading strand of one fork is primarily primed by the lagging strand of the sister replisome (*Aria and Yeeles, 2018*), activation of a single replisome at the origin may be expected to synthesize only the lagging strand in the absence of a leading strand. The lane scan in *Figure 5C* demonstrates that disproportionately high levels of lagging strand products are also not produced in the presence of both Cdt1·MCM$^{2Δ127}$ and wildtype Cdt1·MCM. We conclude that the Mcm2 N-termini of both hexamers are required for efficient origin firing and that unidirectional origin firing is not supported by MCM DHs containing only a single Mcm2 NTE.

## Rad53 sterically inhibits DDK binding to MCM DHs

We have shown that DDK binding to MCM DHs is required for origin activation. We, therefore, asked whether Rad53 might control origin activity by inhibiting DDK binding to MCM DHs. For this we purified recombinant wildtype Rad53, which undergoes autoactivation during overexpression in *E. coli*, or the catalytically dead Rad53$^{D339A}$ mutant, designated Rad53-kd below (*Gilbert et al., 2001*). Indeed, pre-incubation of DDK with Rad53 in the presence of ATP prior to addition of MCM DHs prevented both the binding of DDK to MCM DHs and Mcm4 and −6 phosphorylation by DDK (*Figure 6A*, lanes 4+5). Moreover, Rad53 was able to displace DDK from MCM DHs when DDK binding to MCM DHs preceded addition of Rad53 (lanes 4+7). However, this displacement action of Rad53 was slightly less efficient at disrupting the DDK-MCM DH interaction than the action of preventing DDK recruitment (lanes 5+7). Intriguingly, Rad53-kd also largely inhibited stable binding of

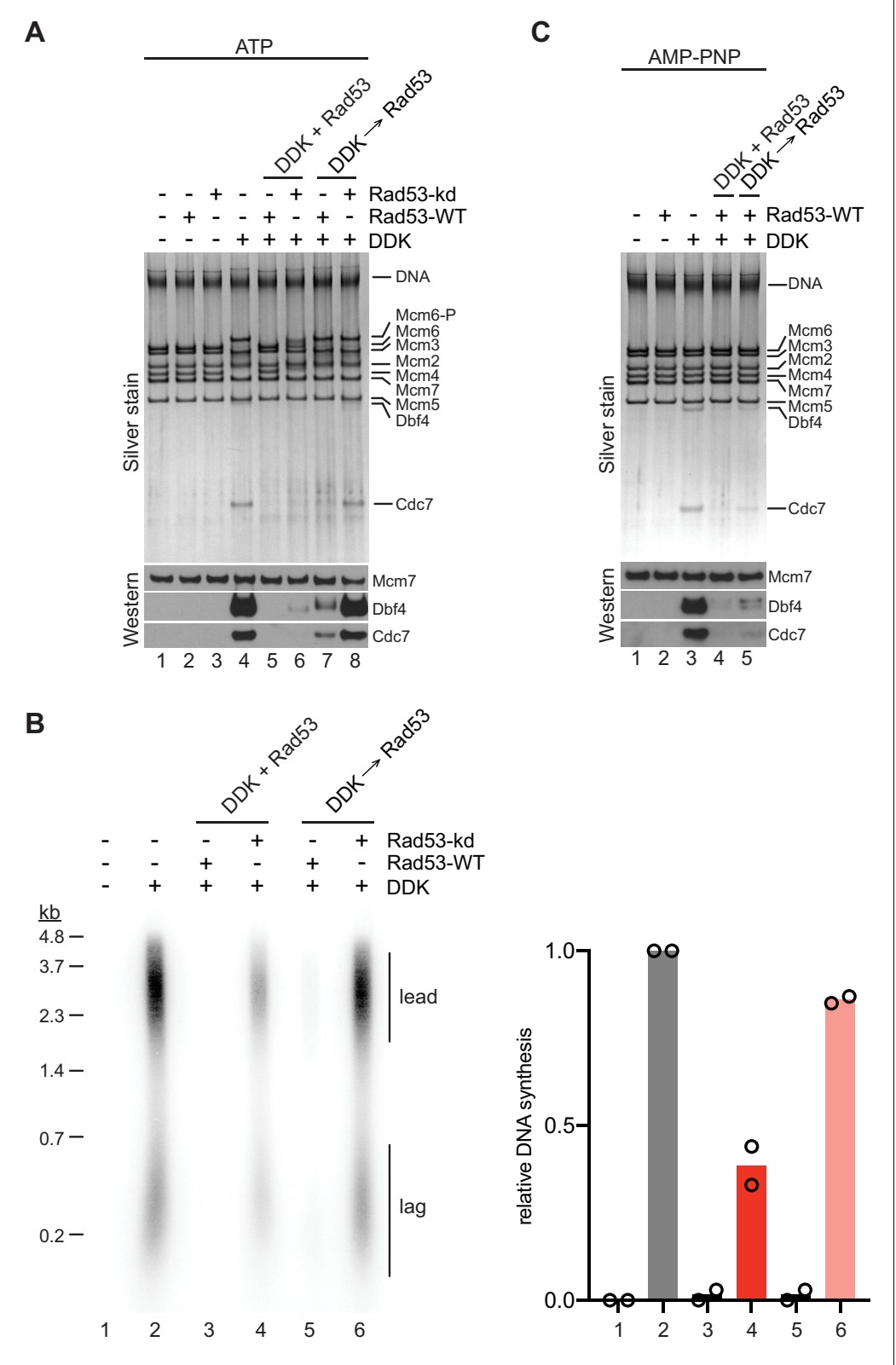

**Figure 6.** Steric inhibition of DDK binding to MCM DHs by Rad53. (**A**) DDK binding to purified MCM DHs was monitored in the presence of ATP and in the absence or presence of Rad53-WT or Rad53-kd, as indicated. In lanes 5+6 DDK and Rad53 were co-incubated in the presence of ATP prior to addition to DNA-bound MCM-7 DHs; in lanes 7+8 DDK was incubated with purified MCM DHs before addition of Rad53. DNA-bound material was analyzed SDS-PAGE and silver stain or western blot as indicated. (**B**) Standard DNA replication reaction using p1017 (4.8 kb) as template. Rad53 and

*Figure 6 continued on next page*

*Figure 6 continued*

DDK were either co-incubated prior to simultaneous addition after the MCM loading step (lanes 3+4) or Rad53 was added after DDK prior to the addition of activation factors (lanes 5+6). Replication products were analyzed by denaturing agarose gel-electrophoresis and autoradiography. The results of two experiment repeats are plotted in the graph on the right. (C) DDK binding to DNA-bound MCM DHs was monitored in the presence of AMP-PNP. DDK and Rad53 were either co-incubated in the presence of AMP-PNP prior to addition to purified DNA-bound MCM DHs (lane 4), or added sequentially to MCM DHs (lane 5) as indicated.

The online version of this article includes the following source data for figure 6:

**Source data 1.** *Figure 6A*, silver stain.
**Source data 2.** *Figure 6A*, immunoblot: Mcm7.
**Source data 3.** *Figure 6A*, immunoblot: Dbf4.
**Source data 4.** *Figure 6A*, immunoblot: Cdc7.
**Source data 5.** *Figure 6B*, autoradiograph.
**Source data 6.** *Figure 6C*, silver stain.
**Source data 7.** *Figure 6C*, immunoblot: Mcm7, Dbf4.
**Source data 8.** *Figure 6C*, immunoblot: Cdc7.

DDK to MCM DHs when co-incubated with DDK prior to addition to Mcm2-7 DHs, demonstrating that DDK phosphorylation by Rad53 is not essential to inhibit stable DDK recruitment to MCM (lanes 4+6). However, some residual DDK binding and significant Mcm4 and −6 phosphorylation occurred in the presence of Rad53-kd, indicating that Rad53-kd-mediated inhibition of DDK is inefficient. Moreover, unlike Rad53-WT, Rad53-kd was unable to displace DDK from MCM DHs when DDK was bound to MCM DHs prior to Rad53-kd addition (lanes 4+8).

Consistent with the DDK binding data we find that Rad53-WT efficiently inhibits origin firing *in vitro* both when pre-incubated with DDK or when added after DDK following MCM loading (*Figure 6B*). In contrast, Rad53-kd was unable to inhibit origin firing when added after DDK to the replication reaction and only partially inhibited origin firing when pre-incubated with DDK prior to addition at the MCM loading step. These observations are consistent with genetic data demonstrating that kinase-dead alleles of Rad53 are deficient in origin inhibition (*Lopes et al., 2001*; *Pellicioli et al., 1999*). We conclude that activated Rad53 inhibits DDK binding to Mcm2-7 DHs, which in turn inhibits Mcm4 and −6 phosphorylation and origin activation.

Our observation that Rad53-WT but not Rad53-kd inhibits DDK binding to MCM DHs and subsequent origin firing is consistent with current models implicating Rad53 phosphorylation of Dbf4 in DDK inhibition. Interestingly, previous studies have indicated that a physical interaction between Rad53 and DDK also contributes to the inhibition of origin activation by the checkpoint (*Chen et al., 2013*; *Duncker et al., 2002*; *Matthews et al., 2014*; *Varrin et al., 2005*). Whether this physical interaction simply mediates Dbf4 phosphorylation by Rad53 or contributes independently to DDK inhibition by Rad53 has not been addressed. Therefore, to determine if Rad53 phosphorylation of DDK is required for the inhibition of DDK binding to MCM DHs we monitored DDK-MCM DH complex formation in the presence of AMP-PNP. As we have shown above, AMP-PNP promotes DDK binding to MCM DHs to the same extent as ATP (*Figure 4F*). Strikingly, the ability of Rad53 to prevent DDK recruitment to MCM DHs or to displace DDK from MCM DHs was undiminished in the presence of AMP-PNP (*Figure 6C*). Thus, DDK phosphorylation by Rad53 is not required to control DDK-MCM DH complex formation.

To address the non-catalytic mechanism by which Rad53 may inhibit DDK, we tested if Rad53 can form a stable complex with DDK that may sequester DDK and thereby inhibit MCM phosphorylation. For this, we monitored Rad53-DDK complex formation by gel-filtration. Intriguingly, Rad53-WT and Rad53-kd exhibit very different elution profiles during gel-filtration chromatography (*Figure 7A*). While Rad53-kd elutes at a volume that is consistent with a monomeric structure, Rad53-WT elutes as a oligomeric complex. Rad53 and its human homolog, Chk2, are known to associate into homo-dimeric complexes during activation by trans-autophosphorylation, suggesting the oligomeric form of Rad53 observed here is a dimer (*Ahn and Prives, 2002*; *Cai et al., 2009*; *Oliver et al., 2006*; *Wybenga-Groot et al., 2014*; *Xu et al., 2002*). However, contrary to our observation, both kinase-dead Rad53 and Chk2 have been shown previously to also form dimers (*Cai et al., 2009*; *Wybenga-Groot et al., 2014*). As these previous studies were carried out with truncated kinase versions, it is

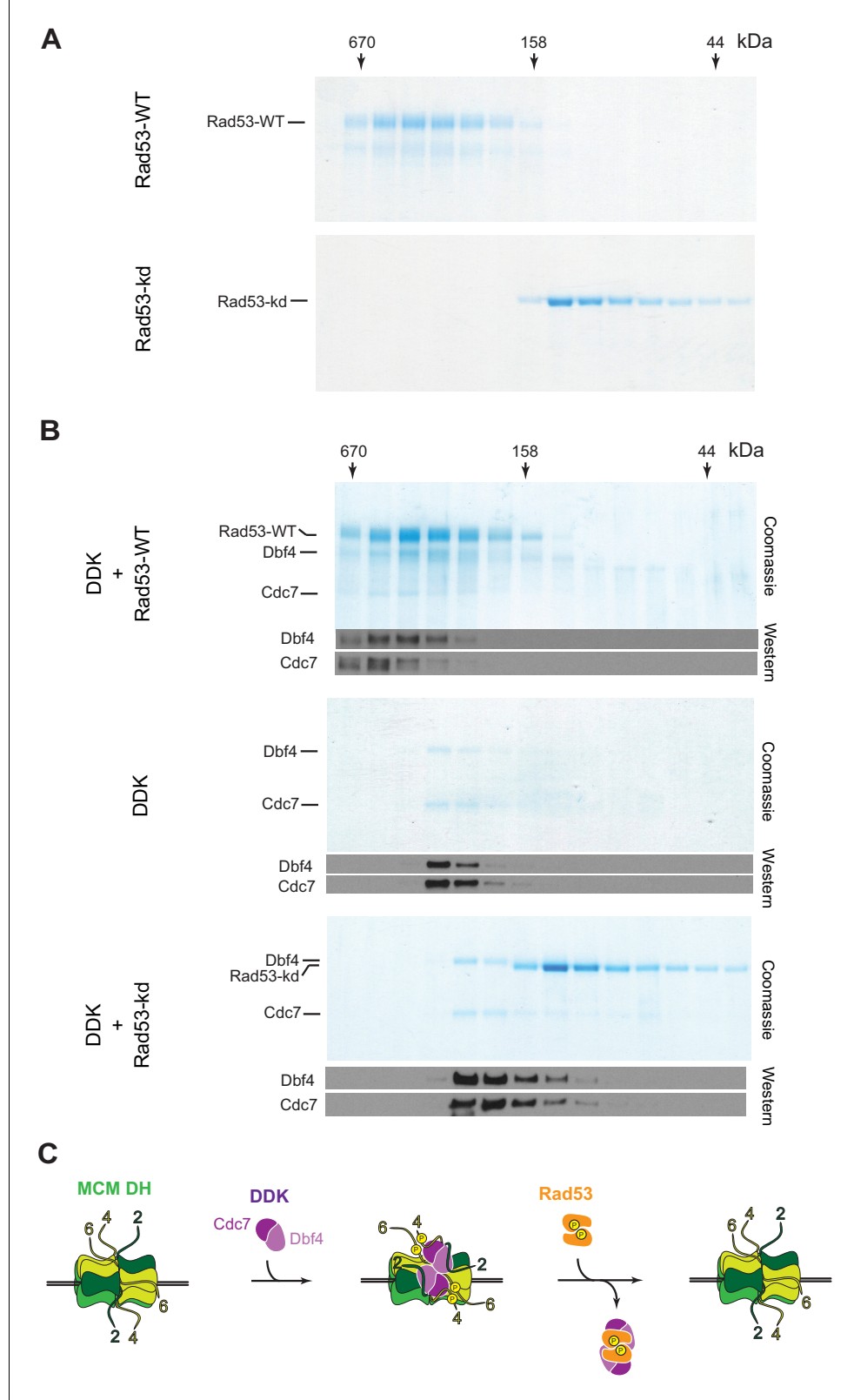

**Figure 7.** Rad53-WT, but not Rad53-kd, can form a stable complex with DDK. (**A**) Gel-filtration analysis of purified Rad53-WT (top) or Rad53-kd (bottom), as indicated. Samples were analyzed by SDS-PAGE and Coomassie stain. (**B**) Gel-filtration analysis of Rad53-WT + DDK (top), DDK alone (center), or Rad53-kd + DDK (bottom). Samples

*Figure 7 continued on next page*

*Figure 7 continued*

were analyzed by SDS-PAGE and Coomassie stain or western blot, as indicated. (**C**) Model illustrating the inhibition of DDK-MCM DH complex formation by competitive binding of activated Rad53 to DDK.

The online version of this article includes the following source data for figure 7:

**Source data 1.** *Figure 7A*, Rad53-WT.
**Source data 2.** *Figure 7A*, Rad53-kd.
**Source data 3.** *Figure 7B*, Rad53-WT + DDK.
**Source data 4.** *Figure 7B*, Rad53-WT + DDK, immunoblot: Dbf4.
**Source data 5.** *Figure 7B*, Rad53-WT + DDK, immunoblot: Cdc7.
**Source data 6.** *Figure 7B*, DDK.
**Source data 7.** *Figure 7B*, DDK, immunoblot: Dbf4, Cdc7.
**Source data 8.** *Figure 7B*, Rad53-kd + DDK.
**Source data 9.** *Figure 7B*, Rad53-kd + DDK, immunoblot: Dbf4, Cdc7.

---

possible that the additional domains present in our full-length Rad53-kd construct affect its dimerization. For example, the isolated Chk2 kinase domain adopts a highly distinct dimer configuration from that of a Chk2 construct spanning the FHA and kinase domain, supporting the notion that domain composition can affect the oligomeric structure of Rad53/Chk2 (*Cai et al., 2009*; *Oliver et al., 2006*). Importantly, Rad53-WT but not Rad53-kd eluted as a stable complex with DDK during gel-filtration (*Figure 7B*). Thus, the ability of Rad53 to bind DDK correlates with its origin inhibition function. We conclude that Rad53 can sterically inhibit DDK binding to MCM DHs, suggesting a novel non-catalytic mechanism for Rad53-dependent origin control.

## Discussion

The head-to-head orientation of the hexamers in the MCM DH requires CMG helicases to pass each other during origin firing (*Douglas et al., 2018*; *Georgescu et al., 2017*). Based on this configuration, it was proposed that an inactive CMG encircling dsDNA blocks the progression of a CMG formed around the opposite hexamer to impose bidirectional origin firing (*Douglas et al., 2018*; *Georgescu et al., 2017*). However, a stalled CMG helicase encircling ssDNA would pose a threat to DNA integrity due to the exposure of the displaced strand at the active CMG. In addition, due to the ability of inactive CMG complexes to slide freely along duplex DNA (*Douglas et al., 2018*; *Wasserman et al., 2019*) we consider it unlikely that an inactive CMG would pose a significant barrier to an opposing replication fork on naked DNA *in vitro*. As an alternative fail-safe mechanism, we propose that simultaneous activation of both MCM hexamers at an origin is controlled by an interdependent mechanism. Our data identify the Mcm2 NTE as a critical component of such a mechanism, as loss of a single Mcm2 N-terminus at a MCM DH inhibits origin activity without inducing unidirectional firing. Mechanistically, we propose that both Mcm2 NTEs in a MCM DH are required for tethering of a pair of DDK molecules to promote symmetric phosphorylation of both hexamers. It will, therefore, be important in future studies to determine the stoichiometry of DDK in the DDK-MCM DH complex. Supporting an interdependent hexamer activation mechanism, recent findings have demonstrated that both MCM hexamers loaded at an origin have to be physically associated to support CMG activation (*Champasa et al., 2019*). Interdependent MCM activation may conceivably be required at the origin melting step, which may involve two CMGs working in opposite direction against each other (*Froelich et al., 2014*; *Langston and O'Donnell, 2019*; *Noguchi et al., 2017*).

Our proposed role of the Mcm2 NTE as a DDK docking site is supported by a previous yeast two-hybrid study that detected a specific pairwise interaction between Dbf4 and the Mcm2 N-terminus (*Ramer et al., 2013*). Although a docking interaction between DDK and the structured NTD of Mcm4 was suggested by *in vitro* studies with purified DDK and Mcm4 (*Sheu and Stillman, 2006*), we find that in the context of the MCM DH this interaction is insufficient to tether DDK in the absence of the Mcm2 NTE. However, our data does not rule out the possibility that DDK docking involves a complex interaction surface comprising critical interactions with multiple MCM subunits. In fact, we show that MCM DHs, but not the single MCM hexamers contained in OCCMs, are stably bound by DDK (*Figure 4B*). This is consistent with previous data demonstrating that MCM DHs are

preferred phosphorylation targets for DDK over single MCM hexamers (*Francis et al., 2009*; *Sun et al., 2014*). Thus, structural determinants in addition to the Mcm2 NTE govern the interaction of DDK with MCM DHs. Moreover, prior phosphorylation of MCM by other kinases has also been shown to promote DDK binding to MCM at origins (*Francis et al., 2009*). In addition, we show that ATP-binding promotes DDK-MCM complex formation. How ATP-binding may promote DDK binding to MCM DHs is not clear, but it is possible that ATP stabilizes an active site conformation in Cdc7 that is more conducive to substrate engagement than the nucleotide-free form, as has been proposed for other protein kinases (*Taylor and Kornev, 2011*). Structural characterization of the DDK·MCM DH complexes isolated here will help define the details of the DDK-MCM interface. Such an analysis may also help resolve the mechanism of Mcm4/6 phosphorylation by DDK, which may occur across the hexamer-hexamer interface or within the hexamer bound by a DDK molecule (*Sun et al., 2014*).

We find that DDK remains stably bound to MCM DHs even after phosphorylation of the Mcm4 and −6 NTEs. DDK phosphorylation of MCM is opposed by dephosphorylation of MCM by the Rif1-PP1 phosphatase complex (*Alver et al., 2017*; *Davé et al., 2014*; *Hiraga et al., 2014*; *Hiraga et al., 2017*; *Mattarocci et al., 2014*). While tethering of DDK to MCM DHs may, therefore, not be essential for maintaining DDK phosphorylation of Mcm4/6 in the absence of Rif1-PP1 *in vitro* (*Figure 3B*, lane six and *Figure 4C*, lane 4), it is likely to promote MCM phosphorylation *in vivo* by shifting the balance between DDK and Rif1-PP1 activity toward DDK. Additionally, as Dbf4 concentrations are limiting for origin activation, retention of DDK at early origins until origin firing may prevent premature recycling of DDK to late origins to establish the replication timing program (*Mantiero et al., 2011*; *Tanaka et al., 2011*). Similarly, retention of DDK at unfired origins has been proposed to restrict DDK targeting of Eco1 until late S phase to control cohesion in the cell cycle (*Seoane and Morgan, 2017*). These mechanisms imply that DDK release from MCM DHs is regulated by origin activation. Since DDK targets MCM DHs specifically, hexamer separation may be sufficient to induce DDK release. Alternatively, tethering of DDK to the long and flexible Mcm2 NTE may allow DDK to remain bound to replisomes. Indeed, the association of DDK with replisomes was shown to link DNA replication with meiotic recombination (*Murakami and Keeney, 2014*). Analogously, DDK bound to replisomes in a mitotic S phase may promote spatio-temporal coordination of DNA replication with other chromosomal processes such as chromatin assembly or DNA repair (*Furuya et al., 2010*; *Gérard et al., 2006*). It will, therefore, be interesting in future experiments to determine the dynamics of the DDK-MCM interaction during DNA replication.

We find that activated Rad53 inhibits DDK binding to MCM DHs by a steric mechanism that is independent of Dbf4 phosphorylation by Rad53 (model *Figure 7C*). Dbf4 contains three evolutionarily conserved sequence motifs, termed N, M, and C that constitute parts of distinct domains separated by flexible linkers (*Hughes et al., 2012*; *Masai and Arai, 2000*). The M and C domains embrace the Cdc7 kinase to mediate Cdc7 activation, while the N motif forms part of an N-terminal BRCT domain that is dispensable for Cdc7 activation (*Almawi et al., 2016*; *Dick et al., 2020*; *Hughes et al., 2012*). Intriguingly, an N-terminal fragment encompassing the BRCT and M domains is required and sufficient to target Dbf4 to MCM complexes loaded at replication origins (*Dowell et al., 1994*; *Francis et al., 2009*), while a physical interaction between Rad53 and the Dbf4 BRCT domain has been proposed to contribute to checkpoint-dependent origin control (*Chen et al., 2013*; *Duncker et al., 2002*; *Matthews et al., 2012*; *Matthews et al., 2014*). Thus, competitive binding of Rad53 to the Dbf4 BRCT domain is likely in part responsible for the inhibition of DDK binding to MCM DHs observed here. The N-terminal region of Dbf4 has also been shown to interact with ORC (*Duncker et al., 2002*). However, consistent with a previous study in whole-cell yeast extracts (*Francis et al., 2009*), we show here that purified MCM DHs can recruit DDK to replication origins independently of ORC and, conversely, that ORC alone is incapable of recruiting DDK to DNA.

Both Rad53-WT and Rad53-kd can inhibit DDK recruitment to MCM DHs, but the block of Mcm4 and −6 phosphorylation and origin activation in the presence of Rad53-kd is incomplete. More strikingly, unlike Rad53-WT, Rad53-kd is completely deficient for displacing DDK from MCM DHs and thereby inhibit origin firing. However, as Rad53-dependent DDK displacement from MCM DHs does not require Rad53 kinase activity, we suggest that the oligomeric state of Rad53, the Rad53 phosphorylation state, or both control the ability of Rad53 to displace DDK from MCM DHs. While Rad53-kd is monomeric in solution, we show that active Rad53-wt forms oligomers, likely dimers, in

solution. We propose that following recruitment to MCM DHs, DDK may form a dimer on the MCM DH surface that is in complex with both Mcm2 NTEs, which we show are required for maintaining DDK at MCM DHs. A DDK dimer bound to a MCM DH may require simultaneous disruption of the interaction of both DDK molecules with MCM for DDK release, explaining the proficiency of dimeric Rad53-WT and, conversely, deficiency of monomeric Rad53-kd to compete DDK off MCM DHs. Following DDK displacement, PP1 would then be able to dephosphorylate the Mcm4 and −6 NTEs and thus inactivate the origin. On the other hand, as DDK is monomeric in solution, both Rad53-WT and Rad53-kd can bind DDK prior to MCM DH binding, but the interaction of Rad53-kd with DDK is unstable and insufficient to block origin firing. Thus, Rad53 would block DDK binding to MCM DHs *in vivo* only after checkpoint-induced activation and dimerization, as expected. To test this hypothesis, it will be important to generate catalytically active mutants of Rad53 that are deficient for stable dimerization.

A Rad53 kinase-independent mechanism for DDK inhibition was unexpected as previous studies have demonstrated that mutation of Rad53 phosphorylation sites in Dbf4 and Sld3 allows late origin firing in the presence of HU or MMS (*Lopez-Mosqueda et al., 2010*; *Zegerman and Diffley, 2010*). It is possible that origin efficiency under this condition will further increase upon disruption of the Rad53-DDK interface. Alternatively, phosphoacceptor-site mutations in Dbf4 may also affect the physical interaction between Dbf4 and Rad53. These possibilities remain to be tested in the future.

# Materials and methods

## Key resources table

| Reagent type (species) or resource | Designation | Source or reference | Identifiers | Additional information |
|---|---|---|---|---|
| Strain, strain background (*Saccharomyces cerevisiae*) | YDR125 | This paper | | Overexpression and purification of FACT (see *Table 1*) |
| Strain, strain background (*Saccharomyces cerevisiae*) | YJF38 | PMID:23474987 | | Overexpression and purification of Cdt1·Mcm2-7$^{WT}$ |
| Strain, strain background (*Saccharomyces cerevisiae*) | YMC5 | PMID:24566988 | | Overexpression and purification of DDK |
| Strain, strain background (*Saccharomyces cerevisiae*) | YSA11 | This paper | | Overexpression and purification of Cdt1·Mcm2-7$^{2-TEV}$ (see *Table 1*) |
| Strain, strain background (*Saccharomyces cerevisiae*) | YSA27 | This paper | | Overexpression and purification of Cdt1·Mcm2-7$^{2-2A}$ (see *Table 1*) |
| Strain, strain background (*Saccharomyces cerevisiae*) | YSA35 | This paper | | Overexpression and purification of DDK (see *Table 1*) |
| Antibody | Anti-Cdc7 (yN-18) (goat polyclonal) | Santa Cruz Biotechnology | Cat. #: sc-11964 RRID:AB_638349 | (1:2000) |
| Antibody | Anti-Dbf4 (yA-16) (goat polyclonal) | Santa Cruz Biotechnology | Cat. #: sc-5706 RRID:AB_637654 | (1:2000) |
| Antibody | Anti-Mcm2 (yN-19) (goat polyclonal) | Santa Cruz Biotechnology | Cat. #: sc-6680 RRID:AB_648843 | (1:2000) |
| Antibody | Anti-Mcm4 (yC-19) (goat polyclonal) | Santa Cruz Biotechnology | Cat. #: sc-6685 RRID:AB_648862 | (1:2000) |
| Antibody | Anti-Mcm5 (yN-19) (goat polyclonal) | Santa Cruz Biotechnology | Cat. #: sc-6687 RRID:AB_648872 | (1:2000) |

*Continued on next page*

*Continued*

| Reagent type (species) or resource | Designation | Source or reference | Identifiers | Additional information |
|---|---|---|---|---|
| Antibody | Anti-Mcm7 (yN-19) (goat polyclonal) | Santa Cruz Biotechnology | Cat. #: sc-6688 RRID:AB_647936 | (1:2000) |
| Antibody | Anti-goat IgG-HRP (mouse monoclonal) | Santa Cruz Biotechnology | Cat. #: sc-6688 RRID:AB_628490 | (1:5000) |
| Recombinant DNA reagent | p79 (pARS1.4.1) | PMID:3281162 | | Plasmid unwinding assay |
| Recombinant DNA reagent | p470 (pARS305) | PMID:27989437 | | Template for MCM loading, phosphorylation, DDK binding, and replication assays |
| Recombinant DNA reagent | p779 (pRS306G-MCM2 /FLAG-MCM3) | PMID:23474987 | | Yeast overexpression of Mcm2 and FLAG-Mcm3, template for Mcm2 modifications |
| Recombinant DNA reagent | p993 (pRS305G-CBP-POB3++) | This paper | | Yeast overexpression of CBP-Pob3 |
| Recombinant DNA reagent | p1000 (pRS306G-SPT16++) | This paper | | Yeast overexpression of Spt16 |
| Recombinant DNA reagent | p1017 (pARS1) | PMID:27989437 | | Template for replication assay |
| Recombinant DNA reagent | p1034 (pRS306G-MCM2-TEV/FLAG-MCM3) | This paper | | Yeast overexpression of Mcm2-TEV and FLAG-Mcm3 |
| Recombinant DNA reagent | p1035 (pet15b-NHP6) | This paper | | Bacterial overexpression of His-Nhp6 |
| Recombinant DNA reagent | p1162 (pRS306G-MCM2-2A/FLAG-MCM3) | This paper | | Yeast overexpression of Mcm2-2A and FLAG-Mcm3 |
| Recombinant DNA reagent | p1220 (pRS305G-CDC7-myc/DBF4-ybbR-FLAG) | This paper | | Yeast overexpression of DDK with Cdc7-myc and Dbf4-ybbR-FLAG |
| Sequence-based reagent | DR772 | IDT | PCR primer | /5PCBio/CCATTATCGAAGGCA |
| Sequence-based reagent | DR2417 | BioSynthesis | PCR primer | TACTGAAATGGTATAC[5-Fluoro-2'-dC]GGTAGATGCATAACGAATTCGCTGCGTAGCATTTGGAG |
| Peptide, recombinant protein | Nap1 (6xHis-Nap1) | PMID:27989437 | | |
| Peptide, recombinant protein | ISW1a (Isw1-3xFLAG) | PMID:27989437 | | |
| Peptide, recombinant protein | ORC (CBP-Orc1) | PMID:23474987 | | |
| Peptide, recombinant protein | Cdc6 | PMID:24566988 | | |
| Peptide, recombinant protein | Cdt1·Mcm2-7$^{WT}$ (3xFLAG-Mcm3) | PMID:23474987 | | |
| Peptide, recombinant protein | Cdt1·Mcm2-7$^{2-TEV}$ (3xFLAG-Mcm3) | This paper | | Purified from *Saccharomyces cerevisiae* cells |
| Peptide, recombinant protein | Cdt1·Mcm2-7$^{2-2A}$ (3xFLAG-Mcm3) | This paper | | Purified from *Saccharomyces cerevisiae* cells |
| Peptide, recombinant protein | DDK (Cdc7-myc) | PMID:24566988 | | |
| Peptide, recombinant protein | DDK (Dbf4-ybbR-3xFLAG/Cdc7 myc) | This paper | | Purified from *Saccharomyces cerevisiae* cells (see Materials and methods) |

*Continued on next page*

*Continued*

| Reagent type (species) or resource | Designation | Source or reference | Identifiers | Additional information |
|---|---|---|---|---|
| Peptide, recombinant protein | Sld3·7 (10xHis-Smt3-Sld3) | PMID:27989437 | | |
| Peptide, recombinant protein | Cdc45 (Cdc45-3xFLAG[int]) | PMID:27989437 | | |
| Peptide, recombinant protein | CDK (Clb5-CBP) | PMID:27989437 | | |
| Peptide, recombinant protein | GINS (Psf1-CBP) | PMID:27989437 | | |
| Peptide, recombinant protein | Pol ε (CBP-Pol2) | PMID:27989437 | | |
| Peptide, recombinant protein | Dpb11 (Dpb11-CBP) | PMID:32341532 | | |
| Peptide, recombinant protein | Sld2 (Sld2-3xFLAG) | PMID:32341532 | | |
| Peptide, recombinant protein | RPA | PMID:27989437 | | |
| Peptide, recombinant protein | Pol α (CBP-Pri1) | PMID:27989437 | | |
| Peptide, recombinant protein | Ctf4 (6xHis-Ctf4) | PMID:27989437 | | |
| Peptide, recombinant protein | RFC (Rfc1-FLAG-HAT) | PMID:27989437 | | |
| Peptide, recombinant protein | PCNA (6xHis-PCNA) | PMID:27989437 | | |
| Peptide, recombinant protein | Pol δ (GST-Pol3) | PMID:27989437 | | |
| Peptide, recombinant protein | Csm3·Tof1 (CBP-Csm3) | PMID:32341532 | | |
| Peptide, recombinant protein | Mrc1 (Mrc1-3xFLAG) | PMID:32341532 | | |
| Peptide, recombinant protein | Mcm10 (6xHis-Mcm10) | PMID:24566988 | | |
| Peptide, recombinant protein | Top1 (Top1-CBP) | PMID:27989437 | | |
| Peptide, recombinant protein | Top2 (CBP-Top2) | PMID:27989437 | | |
| Peptide, recombinant protein | Nhp6 (6xHis-Nhp6) | This paper | | Purified from *E. coli* BL21-Codon Plus (DE3)-RIL cells (see Materials and methods) |
| Peptide, recombinant protein | FACT (CBP-Pob3) | This paper | | Purified from *Saccharomyces cerevisiae* cells (see Materials and methods) |
| Peptide, recombinant protein | Rad53 (6xHis-Rad53) | PMID:32341532 | | |
| Peptide, recombinant protein | Rad53[D339A] (6xHis-Rad53[D339A]) | PMID:32341532 | | |
| Peptide, recombinant protein | HpaII methyltransferase | NEB | Cat. #: M0214S | |
| Commercial assay or kit | SilverQuest Silver Staining Kit | Invitrogen (ThermoFisher) | Cat. #: LC6070 | |
| Chemical compound, drug | ATP | Thermo Scientific (Thermo Fisher) | Cat. #: R1441 | |

*Continued on next page*

*Continued*

| Reagent type (species) or resource | Designation | Source or reference | Identifiers | Additional information |
|---|---|---|---|---|
| Chemical compound, drug | ATPγS | Roche (MilliporeSigma) | Cat. #: 11162306001 | |
| Chemical compound, drug | AMP-PNP | Roche (MilliporeSigma) | Cat. #: 10102547001 | |
| Software, algorithm | ImageJ software | ImageJ (http://imagej.nih.gov/ij/) | RRID:SCR_003070 | |
| Software, algorithm | GraphPad Prism software | GraphPad Prism (https://graphpad.com) | RRID:SCR_015807 | |

## Protein purification

Proteins were purified as described previously, unless specified below (*Devbhandari et al., 2017*; *Devbhandari and Remus, 2020*).

### DDK

Two DDK variants, harboring either a removable C-terminal TAP[tcp] tag (*Gros et al., 2014*) or a ybbR-FLAG tag on Dbf4 were used interchangeably. Both variants behave identically and are fully proficient for DNA replication *in vitro*.

The ybbR-FLAG-tagged DDK was purified from strain YSA35. Cells were grown in 48L YP/2 % glycerol/2 % lactic acid pH 5.5 (YPLG) at 30°C to a density of $2 \times 10^7$ cells/mL. Protein expression was induced with 2 % galactose for 4 hr. Cells were collected by centrifugation, washed with 25 mM HEPES-KOH pH 7.6/1 M sorbitol and resuspended in 0.5 volumes of buffer A (45 mM HEPES-KOH pH 7.6/0.02 % NP-40 substitute/10 % glycerol)/100 mM NaCl/1 mM DTT/1 x protease inhibitor cocktail (Pierce). The cell suspension was pipetted dropwise into liquid nitrogen to generate frozen popcorn and stored in −80°C. Cells were lysed by crushing the popcorn in a Spex freezer mill, using 10 cycles of 2 min run + 1 min cooldown at 15 CPS. Resulting whole cell lysate was thawed and supplemented with 1 volume of 45 mM buffer A/100 mM NaCl/1 mM DTT/1 x protease inhibitor cocktail. 5 M NaCl was added to the lysate to a final concentration to 300 mM. After 20 min of gentle agitation at 4°C, cell lysate was centrifuged in a T-647.5 rotor (Thermo Fisher) at 40,000 rpm for 1 hr at 4°C. The clear soluble phase was recovered and DDK pulled down with 1 mL packed FLAG affinity agarose beads (Sigma) for 4 hr at 4°C with gentle rocking. Beads were collected by centrifugation and washed with 10 volumes of buffer A/300 mM NaCl/1 mM DTT. Beads were resuspended in 1 volume of buffer A/300 mM NaCl/2 mM MnCl$_2$/1 mM DTT and incubated with λ protein phosphatase (NEB) at 50 U/mL for 1 hr at 23°C with agitation. Beads were collected and protein eluted in five volumes of buffer A/300 mM NaCl/1 mM DTT supplemented with 0.25 mg/mL 3xFLAG peptide. Eluates were analyzed by SDS-PAGE. The FLAG pulldown was repeated until DDK was depleted from the extract. Fractions containing DDK were pooled, and the volume reduced to 0.5 mL using an Amicon spin concentrator (Millipore). The pooled, concentrated eluate was fractionated on a 24 mL Superdex 200 Increase 10/300 GL (GE Healthcare) gel filtration column in buffer A/300 mM NaCl/1 mM DTT. Fractions were analyzed by SDS-PAGE and peak fractions containing DDK were pooled and concentrated using Amicon spin concentrator before dialysis against buffer A/100 mM KOAc/2 mM β-mercaptoethanol. The concentration of the purified DDK was determined by SDS-PAGE and Coomassie stain using BSA standards. Purified DDK was stored in aliquots a −80°C.

### Nhp6

Nhp6 was expressed as a N-terminal 6x His-tag fusion protein in *E. coli* BL21-CodonPlus (DE3)-RIL cells (Agilent). A colony of cells freshly transformed with plasmid p1035 was grown in 3 L of LB supplemented with 50 µg/mL ampicillin and 34 µg/mL chloramphenicol at 37°C. At OD$_{600}$ ~0.6, 1 mM IPTG was added and the temperature reduced to 4°C. After 1 hr, the temperature was raised to 20°C and the cells were incubated for an additional 16 hr. Cells were collected by centrifugation, rinsed twice with dH$_2$O, once with buffer B (50 mM Tris-HCl pH 7.5/1 mM EDTA/10 % glycerol/10 mM benzamidine/150 mM NaCl), and resuspended in buffer B supplemented with 1x protease inhibitor cocktail (Pierce) and 1 mM DTT. Cells were lysed by addition of 10 mg lysozyme (Thermo Scientific)

**Table 1.** Yeast strains.

| Strain name | Genotype | Purpose |
|---|---|---|
| YDR125 | W303-1a *MATa ade2-1 trp1-1 can1-100 pep4::kanMX bar::hphNAT1 (hygromycinB) his3-11,15::P/Gal 1,10-GAL4 (HIS3) leu2-3,112::P/Gal 1,10-CBP-POB3++ (LEU2) ura3-1::P/Gal 1,10-SPT16++ (URA3)* | Overexpression and purification of FACT |
| YSA11 | W303-1a *MATa ade2-1 can1-100 pep4::kanMX bar1::hphNAT1 (hygromycinB) his3-11,15::GAL4-P/Gal1,10-CDT1 (HIS3) trp1-1::MCM5-P/Gal1,10-MCM4 (TRP1) leu2-3,112::MCM7-P/Gal1,10-MCM6 (LEU2) ura3-1::MCM2-TEV-P/Gal1,10-FLAG-MCM3 (URA3)* | Overexpression and purification of Cdt1·Mcm2-7$^{2-TEV}$ |
| YSA27 | W303-1a *MATa ade2-1 can1-100 pep4::kanMX bar1::hphNAT1 (hygromycinB) his3-11,15::GAL4-P/Gal1,10-CDT1 (HIS3) trp1-1::MCM5-P/Gal1,10-MCM4 (TRP1) leu2-3,112::MCM7-P/Gal1,10-MCM6 (LEU2) ura3-1::MCM2-2A-P/Gal1,10-FLAG-MCM3 (URA3)* | Overexpression and purification of Cdt1·Mcm2-7$^{2-2A}$ |
| YSA35 | W303-1a *MATa ade2-1 ura3-1 trp1-1 can1-100 pep4::kanMX bar::hphNAT1 (hygromycinB) his3-11,15::P/Gal 1,10-GAL4 (HIS3) leu2-3,112::CDC7-myc-P/Gal 1,10-DBF4-ybbR-FLAG (LEU2)* | Overexpression and purification of DDK |

and incubation for 30 mins at 4°C followed by sonication. The clear, soluble phase was isolated after centrifugation of the whole-cell lysate in a T-647.5 rotor (Thermo Fisher) at 40,000 rpm for 30 min at 4°C. Nhp6 was pulled down with 0.5 mL packed Ni-NTA agarose beads (Qiagen) for 3 hr at 4°C with gentle agitation. Beads were collected and washed with 20 volumes of Buffer B/1 mM DTT. Protein was eluted with five volumes of buffer B/1 mM DTT/100 mM imidazole, and eluates were analyzed by SDS-PAGE. Eluate fractions containing Nhp6 were pooled and the volume reduced to 0.5 ml using an Amicon spin concentrator (Millipore). The pooled concentrate was fractionated by gel filtration chromatography using a Superdex 200 10/300 GL (GE Healthcare) column in buffer B/1 mM DTT. Elution fractions were analyzed by SDS-PAGE and Nhp6 peak fractions were pooled and concentrated using a spin concentrator. Purified Nhp6 was aliquoted, snap-frozen, and stored at −80°C.

## FACT

FACT complex was purified after overexpression in yeast cells harboring codon-optimized copies of *SPT16* and N-terminally CBP-tagged *POB3* under control of the GAL1,10 promoter (strain YDR 125). Cells were grown in 12 L YPLG at 30°C up to a density of $2 \times 10^7$ cells/mL. Protein expression was induced with 2 % galactose for 4 hr. Cells were collected by centrifugation, washed with 25 mM HEPES-KOH pH 7.6/1 M sorbitol, and resuspended in 0.5 volumes of Buffer C (25 mM Tris-HCl pH 7.5/0.02 % NP-40 substitute/10 % glycerol)/100 mM NaCl/1 mM DTT/1 x protease inhibitor cocktail (Pierce). The cell suspension was pipetted dropwise into liquid nitrogen to generate frozen popcorn and stored at −80°C. Cells were lysed by crushing in a Spex freezer mill with 10 cycles of 2 min run and 1 min cooldown at 15 CPS. Thawed whole cell lysate was supplemented with 1 volume of Buffer C/100 mM NaCl/1 mM DTT/1 x protease inhibitor cocktail and the final concentration of NaCl adjusted to 300 mM using a 5 M NaCl stock solution. After 20 min of gentle agitation at 4°C, the cell lysate was centrifuged in a T-647.5 rotor (Thermo Fisher) at 40,000 rpm for 1 hr at 4°C. The clear soluble phase was recovered and supplemented with 2 mM CaCl$_2$. FACT was pulled down from the extract with 0.5 mL packed calmodulin affinity resin (Agilent) for 4 hr at 4°C with gentle rocking. Resin was collected by centrifugation and washed with 10 volumes of Buffer C/300 mM NaCl/2 mM CaCl$_2$/1 mM DTT. Protein was eluted from the resin with 7 volumes of Buffer C/300 mM NaCl/1 mM EDTA/2 mM EGTA/1 mM DTT, and eluates were analyzed by SDS-PAGE and Coomassie stain. The calmodulin pulldown was repeated until FACT was depleted from the extract. Fractions containing FACT were pooled and incubated for 16 hr at 4°C with 400 µg TEV protease to remove the CBP tag. The digest was diluted threefold with buffer D (25 mM Tris-HCl pH 7.5/1 mM EDTA/10 % glycerol)/1 mM DTT to reduce the final NaCl concentration to 100 mM, and fractionated on a MonoQ 5/50 GL (GE Healthcare) column in buffer D/1 mM DTT using a gradient of 0.1–1 M NaCl over 20 column volumes. Fractions containing FACT were pooled and concentrated to a volume of 0.5 ml using an Amicon spin concentrator (Millipore). The concentrate was gel-filtered on a 24 ml Superdex 200 10/300 GL (GE Healthcare) column equilibrated in 25 mM HEPES-KOH pH 7.5/1 mM EDTA/10 % glycerol/300 mM KOAc/1 mM DTT. Elution fractions were analyzed by SDS-PAGE and Coomassie stain. FACT-containing peak fractions were pooled, spin-concentrated, aliquoted, snap-frozen and stored at −80°C.

## Cdt1·MCM$^{2\Delta127}$

Purified Cdt1·Mcm2-7$^{2\text{-TEV}}$ complex was supplemented with 22.5-fold molar excess of TEV protease and incubated for 1 hr at 30℃. The reaction was fractionated on a Superdex 200 Increase 10/300 GL (GE Healthcare) gel filtration column in 45 mM HEPES-KOH pH 7.6/5 mM Mg(OAc)$_2$ / 0.02 % NP-40 substitute/10 % glycerol/100 mM KOAc/1 mM ATP/1 mM DTT, and Cdt1·MCM$^{2\Delta127}$-containing peak fractions pooled, aliquoted, snap-frozen in liquid nitrogen and stored at −80℃.

## Gel-filtration analysis of Rad53-DDK complex

1.2 µM purified DDK was incubated with 1.4 µM purified Rad53 variant in 32 mM HEPES-KOH pH 7.6/200 mM KOAc/10 mM Mg(OAc)$_2$ / 0.5 mM EDTA/0.5 mM EGTA/0.01 % NP-40 substitute/10 % glycerol/1 mM AMP-PNP/1 mM DTT for 30 min at 30℃. The sample was fractionated on a Superdex 200 Increase 3.2/300 (GE Healthcare) gel filtration column in 25 mM HEPES-KOH pH 7.6/10 mM Mg (OAc)$_2$/0.02 % NP-40 substitute/5 % glycerol/185 mM KOAc/1 mM DTT/1 mM AMP-PNP. Elution fractions were analyzed by SDS-PAGE and Coomassie stain or western blotting.

## DNA templates

### DNA beads

MCM loading, MCM phosphorylation, and MCM-DDK binding assays were performed on a linear 3 kbp ARS305-containing DNA covalently linked on one end to HpaII methyltransferase (M.HpaII) and immobilized on paramagnetic beads via a 5′ photocleavable biotin on the other end. The template was PCR-amplified from p470 using oligo DR772, which contains a photocleavable 5′ biotin moiety, and oligo DR2417, which contains a M.HpaII-binding sequence modified with 5-fluoro-2′-deoxycyti-dine (BioSynthesis). The purified PCR product was coupled to Dynabeads M280 streptavidin magnetic beads (Invitrogen). M.HpaII (NEB) was conjugated to bead-bound DNA in 50 mM Tris-HCl pH 7.5, 10 mM EDTA, 100 µM SAM at a ratio of 4 units M.HpaII per 90 fmol of DNA for 16 hr at 37℃ with agitation. M.HpaII-conjugated bead-bound DNA was washed and stored in 10 mM HEPES-KOH pH 7.6/50 mM KOAc/1 mM DTT at 4℃.

### Plasmids

The plasmid unwinding assay was performed on circular 3 kbp ARS1-containing p79, while the *in vitro* DNA replication assays were performed on ARS1-containing p1017 (4.8 kbp) or ARS305-containing p470 (10 kbp) DNA. Plasmid DNAs were initially isolated using a maxiprep kit (Qiagen). To remove nicked plasmid species, purified plasmid DNA was fractionated on a 10–40% sucrose gradient in 20 mM Tris-HCl pH 7.5/1 mM EDTA/1M NaCl using an AH-629 swinging bucket rotor (Thermo Scientific) at 27,000 rpm for 20 hr at 20℃. 0.5 ml fractions were collected and analyzed by agarose gel-electrophoresis in the absence of ethidium bromide. The gel was stained post-run with ethidium bromide. Supercoiled DNA-containing fractions were pooled, dialyzed against 10 mM Tris pH 7.5/2 mM EDTA, concentrated using an Amicon spin concentrator (Millipore) to 1 to 2 mg/ml, and stored in aliquots at −20℃.

To generate chromatinized templates for DNA replication assays, 1.5 µg Nap1, 0.5 µg Histone octamer, 0.2 µg Isw1a, and 0.8 pmol ORC were mixed and incubated in 10 mM HEPES-KOH pH 7.5/50 mM KCl/5 mM MgCl$_2$/0.5 mM EGTA/10 % glycerol/0.1 mg/mL BSA for 30 min at 4℃. 0.5 µg of purified supercoiled plasmid DNA was subsequently added along with 3 mM ATP, 20 mM creatine phosphate, and 50 ng/µL creatine kinase in a total volume of 10 µL for 1 hr at 30℃. The chromatin template was immediately used for *in vitro* replication.

## MCM loading assay

MCM loading reactions were carried out with agitation for 30 min at 30℃ in a 40 µL reaction volume in 25 mM HEPES-KOH pH 7.6/10 mM Mg(OAc)$_2$/0.02 % NP-40 substitute/5 % glycerol/100 mM KOAc/1 mM DTT/5 mM ATP or ATPγS, using 88 nM ORC, 86 nM Cdc6, 420 nM Cdt1·Mcm2-7 (wild-type or mutant as indicated), and 0.3 µg of bead-bound DNA. After the reaction, beads were magnetically separated from the supernatant, and washed once with Wash Buffer (45 mM HEPES-KOH pH 7.6/5 mM Mg(OAc)$_2$/1 mM EDTA/1 mM EGTA/0.02 % NP-40 substitute/10 % glycerol)/300 mM KOAc, once with Wash Buffer/500 mM NaCl, and once with Binding Buffer (25 mM HEPES-KOH pH 7.6/10 mM Mg(OAc)$_2$/0.02 % NP-40 substitute/5 % glycerol/100 mM KOAc). Beads were

resuspended in Binding Buffer/1 mM DTT and supplemented with either 9.45 µM TEV protease (a 22.5-fold molar excess over Cdt1·Mcm2-7) or an equal volume of TEV protease storage buffer (50 mM Tris-HCl pH 7.5/1 mM EDTA/10 % glycerol/100 mM NaCl/1 mM DTT) as a mock control in a total volume of 40 µL. The reactions were incubated for 1 hr at 30°C with agitation. Beads were magnetically separated from the supernatant and washed once with Wash Buffer/300 mM KOAc, once with Wash Buffer/500 mM NaCl, and once with Binding Buffer. Beads were resuspended in 20 µL Wash Buffer/50 mM KOAc/1 mM DTT, and the DNA eluted from the beads by exposure to UV$_{312\text{ nm}}$ for 10 min using a hand-held UV lamp. The supernatant, containing the DNA and DNA-bound proteins, was analyzed by SDS-PAGE followed by silver staining and immunoblot. Replicate experiments were performed two to three times to ensure reproducibility.

## MCM phosphorylation assay

MCM loading was carried out as described above. Following the TEV protease or mock cleavage and wash steps, beads were resuspended in Binding Buffer/5 mM ATP/1 mM DTT and supplemented with purified DDK at 150 nM or indicated concentrations in a total volume of 40 µL. The reaction was incubated for 20 min at 30°C with agitation. Beads were magnetically separated from the supernatant, and washed once with Wash Buffer/300 mM KOAc, once with Wash Buffer/500 mM NaCl, and once with Binding Buffer. Beads were resuspended in 20 µL Wash Buffer/50 mM KOAc/1 mM DTT, and the DNA was eluted from the beads by exposure to UV$_{312\text{ nm}}$ for 10 min. The supernatant, containing the DNA and DNA-bound proteins, was analyzed by SDS-PAGE followed by silver staining and immunoblot.

## DDK-MCM DH binding assay

MCM loading was carried out as described above. For *Figure 4B*, the 500 mM NaCl wash was omitted for samples in lanes 2 to 8 and mock TEV protease cleavage was omitted for all samples. Following the TEV protease or mock cleavage and wash steps, beads were resuspended in Binding Buffer/5 mM ATPγS/1 mM DTT and supplemented with purified DDK at 150 nM or indicated DDK concentrations in a total volume of 40 µL. For *Figures 4F*, *6A and C*, ATPγS was substituted with the indicated ATP analog. Binding reactions were incubated for 20 min at 30°C with agitation. For *Figure 6A and C*, the indicated Rad53 variant was either pre-mixed with DDK or added sequentially. To pre-mix, 250 nM Rad53 was incubated with 150 nM DDK in 10 mM Mg(OAc)$_2$ and 5 mM ATP or AMP-PNP for 20 min at 30°C before adding to the resuspended beads. For sequential addition, the binding reaction was carried out as above for 10 min instead of 20 min, at which point 250 nM Rad53 was added and the reaction carried out for another 10 min. Beads were magnetically separated from the supernatant and washed once with Wash Buffer/100 mM KOAc. For *Figure 4C*, another round of TEV or mock cleavage was performed in Binding Buffer/5 mM ATPγS/1 mM DTT for 1 hr at 30°C followed by a wash with Wash Buffer/100 mM KOAc. For *Figure 4E*, beads were washed in Wash Buffer with the indicated salt concentrations. Beads were resuspended in 20 µL Wash Buffer/50 mM KOAc/1 mM DTT, and the DNA was eluted from the beads by exposure to UV$_{312\text{ nm}}$ for 10 min. The supernatant, containing the DNA and DNA-bound proteins, was analyzed by SDS-PAGE followed by silver staining and immunoblot. Experiments were repeated two to three times to ensure reproducibility.

## Plasmid unwinding assay

0.5 µg of supercoiled p79 plasmid DNA was first relaxed with 100 nM purified Top1 in 25 mM HEPES-KOH pH 7.6/10 mM Mg(OAc)$_2$ / 0.02 % NP-40 substitute/5 % glycerol/100 mM KOAc/1 mM DTT in a reaction volume of 10 µL for 1 hr at 30°C. MCM loading was subsequently carried out in 25 mM HEPES-KOH pH 7.6/10 mM Mg(OAc)$_2$/0.02 % NP-40 substitute/5 % glycerol/100 mM KOAc/1 mM DTT/3.5 mM ATP using 65 nM ORC, 104 nM Cdc6, 130 nM Cdt1·Mcm2-7, 20 mM creatine phosphate, and 50 ng/µL creatine kinase in a reaction volume of 20 µL for 30 min at 30°C. Where indicated, 2.9 µM TEV protease was added (a 22.5-fold molar excess over Cdt1·Mcm2-7) for 1 hr at 30°C. 60 nM DDK was then added for 20 min at 30°C. For CMG assembly and activation the reaction was supplemented with 34 nM CDK, 0.5 mg BSA, 40 nM Sld3·7, 40 nM Cdc45, 50 nM GINS, 34 nM Pol ε, 54 nM Dpb11, 40 nM Sld2, 100 nM RPA, 20 nM Top1 and 14 nM Mcm10 in a total volume of 50 µL with the salt concentration adjusted to 185 mM KOAc for 30 min at 30°C. The reaction was

quenched with 17 mM EDTA, 0.2 % SDS, and 0.8 U Proteinase K (NEB) for 30 min at 37°C. DNA was purified by phenol:chloroform extraction and centrifugation through a G-25 spin column (GE Healthcare). Samples were analyzed by agarose gel-electrophoresis and post-run ethidium-bromide stain. The experiments were repeated twice.

### *In vitro* DNA replication assay

MCM loading was performed for 30 min at 30°C in a reaction volume of 20 µL including 40 nM ORC, 64 nM Cdc6, 80 nM Cdt1·Mcm2-7, 0.5 µg supercoiled plasmid DNA and a reaction buffer containing 25 mM HEPES-KOH pH 7.6/10 mM Mg(OAc)$_2$/0.02 % NP-40 substitute/5 % glycerol/100 mM KOAc/ 1 mM DTT/3.5 mM ATP/20 mM creatine phosphate/50 ng/µL creatine kinase. Where indicated, 1.8 µM TEV protease or an equivalent volume of TEV storage buffer was added for 1 hr at 30°C. 60–140 nM DDK was then added for 20 min at 30°C. For *Figures 6B*, 160 nM Rad53 variant was either pre-mixed with 140 nM DDK or added sequentially as in the DDK-MCM DH binding assay. Origin firing was then induced by supplementing the reaction with 0.5 mg BSA, 40 nM Sld3·7, 40 nM Cdc45, 35 nM CDK, 50 nM GINS, 34 nM Pol ε, 30 nM Dpb11, 40 nM Sld2, 135 nM RPA, 40 nM Pol α, 35 nM Ctf4, 40 nM RFC, 35 nM PCNA, 4 nM Pol δ, 14 nM Csm3-Tof1, 14 nM Mrc1, 20 nM Top1, 15 nM Top2, 7 nM Mcm10, 122 µM each NTP, 40 µM each dNTP, and trace amount of α-$^{32}$P-dATP or α-$^{32}$P-dCTP in a total volume of 50 µL with the salt concentration adjusted to 185 mM KOAc for 30 min at 30°C. Reactions on chromatin templates were additionally supplemented with 2 µM Nhp6 and 200 nM FACT. The reactions were quenched with 17 mM EDTA, 0.2% SDS, and 0.8U Proteinase K (NEB) for 30 min at 37°C. DNA was isolated by phenol:chloroform extraction and centrifugation through a G-25 spin column (GE Healthcare). Samples were run on 0.8 % denaturing agarose gel in 30 mM NaOH/2 mM EDTA, dried, and exposed to a phosphorimager screen. Images were scanned on a Typhoon FLA-9500 and analyzed with ImageJ.

## Acknowledgements

This work was supported by NIGMS grants R01-GM127428 and R01-GM107239 (DR).

## Additional information

### Funding

| Funder | Grant reference number | Author |
| --- | --- | --- |
| National Institute of General Medical Sciences | R01-GM127428 | Dirk Remus |
| National Institute of General Medical Sciences | R01-GM107239 | Dirk Remus |

The funders had no role in study design, data collection and interpretation, or the decision to submit the work for publication.

### Author contributions

Syafiq Abd Wahab, Conceptualization, Investigation, Methodology, Writing - review and editing; Dirk Remus, Conceptualization, Supervision, Funding acquisition, Validation, Writing - original draft, Project administration, Writing - review and editing

### Author ORCIDs

Syafiq Abd Wahab (iD) https://orcid.org/0000-0002-3611-2552
Dirk Remus (iD) https://orcid.org/0000-0002-5155-181X

### Decision letter and Author response

Decision letter https://doi.org/10.7554/eLife.58571.sa1
Author response https://doi.org/10.7554/eLife.58571.sa2

## Additional files

### Supplementary files
• Transparent reporting form

### Data availability
All data are included in the manuscript.

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
