## [Decision Letter]

**Acceptance summary:**

The paper reports an interesting role for the DDK (Cdc7-Dbf4) protein kinase in activating the Mcm2-7 double hexamer after formation of pre-Replictive Complexes are formed on origins of DNA replication. The activity is mediated by DDK binding to an essential region within the Mcm2 amino-terminus and DDK then phosphorylates Mcm4 and Mcm6 subunits. The paper also demonstrates a new role for the checkpoint kinase Rad53 that blocks DDK activation of the Mcm2-7 double hexamer. The observations point to a new mechanism for control of DNA replication initiation.

**Decision letter after peer review:**

Thank you for submitting your article "Antagonistic control of DDK binding to licensed replication origins by Mcm2 and Rad53" for consideration by *eLife*. Your article has been reviewed by three peer reviewers, including Bruce Stillman as the Reviewing Editor and Reviewer #1, and the evaluation has been overseen by a Reviewing Editor and Kevin Struhl as the Senior Editor.

The reviewers have discussed the reviews with one another and the Reviewing Editor has drafted this decision to help you prepare a revised submission.

Summary:

The authors have studied how the DDK (Cdc7-Dbf4) protein kinase activates the initiation of DNA replication. They show, using an Mcm2 subunit in which the amino-terminus can be removed, that this region of Mcm2 is not required for Mcm2-7 double hexamer (DH) formation, but is required for DNA replication. The Mcm2 amino terminus recruits the DDK and promotes phosphorylation of Mcm4 and Mcm6 subunits, but after that it is not required for the Sld3-dependent step in the initiation of DNA replication. Finally, the authors address the role of Rad53 in the inactivation of DDK phosphorylation. They find that Rad53 inhibits DDK binding to loaded Mcm2-7 double hexamers or that it promotes DDK removal from the Mcm2-7 DH and find that this inhibition is lessened by mutating the Rad53 kinase domain but does not require ATP hydrolysis or phosphorylation of Mcm2-7 by DDK or Rad53.

This is a nice paper whose strength are the studies identifying the N-terminal extension of Mcm2 as a DDK binding site on the Mcm2-7 double hexamer. The evidence that removing this binding site inhibits Mcm2-7 phosphorylation, DNA unwinding, and DNA replication initiation is robust and compelling. The use of a protease cleavable form of the Mcm2 protein is a nice touch that allows the removal of the tail at various times in any reaction. This provides clear evidence that the 1-127 region of Mcm2 is important at the DDK step of the initiation reaction. Although this finding does not eliminate additional roles for this region that act later during replication, the authors point out that the distribution of products is not changed suggestion no role during elongation. It is interesting that there is relatively little effect of including chromatinized DNA in the reaction, given that part of what is removed is the binding site for histones.

The last part of the work regarding the involvement of Rad53, and specifically that Rad53 dimerization is needed for DDK displacement by Rad53, is more speculative and would need more evidence to validate the model presented in Figure 7. Thus, toning down the discussion on this point would be valuable.

The revised paper does not require new experimental data, but addressing the issues raised below in a revised manuscript would strengthen the report.

Revisions:

1) The level of DDK required for maximal binding to the Mcm2-7 DH and the phosphorylation of Mcm4 and Mcm6 is 75nM under the conditions employed, but Figure 5—figure supplement 1 shows that 300nM of DDK does not saturate the amount of DNA synthesis. Perhaps the authors could comment on this difference. Of course, DDK might perform multiple functions in the initiation of DNA replication.

2) In Figure 1, does the NTE truncation affect MCM loading (as well as DH stability)? This issue is actually addressed in Figure 5B, but the issue would usefully be covered at the start, either by including data in this figure or by making reference in the text to where the experiment will be presented.

3) Figure 3: Why does the efficiency of replication appear increased when TEV protease is added to MCM2-wt (Figure 3B)? Also, in Figure 3B, it would be useful to show a “-TEV protease” negative control within the Mcm2-TEV set of reactions.

4) Figure 4: Figure 4C: What concentration of DDK was used (to allow comparison with parts A and B)? Figure 4E: A band shift suggests that Dbf4 seems to be phosphorylated/modified when ATP is added. Could the authors explain, and discuss possible reasons and interpretation?

Could the authors explain in the text for what reason ATP/an ATP analogue might be needed to promote full levels of DDK association?

5) Figure 5: Figure 5C: The text states that there is no evidence of asymmetric or uni-directional replication coming from the activation of a single hexamer, but the reasons given for this interpretation are difficult to understand. This conclusion needs to be better explained.

6) The weakest aspect of the paper revolves around the model that DDK bound Mcm2 is required for both of the two Mcm2-7 complexes for efficient initiation to occur. Although the authors propose a model that the 1:1 complex is inactive, the data is not sufficiently different from what would be expected if this complex were just slightly less active (e.g. 2-fold). More importantly, the model of that DDK bound to one kinase phosphorylates the tails of the other kinase has no support in the paper and should be removed or additional experiments added. If this trans model is right, then there are clear experiments to demonstrate this is the case that are well within the authors' abilities. They should be added to support the model or the model should be left out of the paper so that if or when they demonstrate this interesting idea it can be a clear new result. This is not an essential add but if such experiments are not added then the model should be removed. The authors would need more replicates of Figure 6 C, D, and E to be a convincing quantitative argument. Even then, there are clearly other explanations for the data.

7) Figure 6: It is shown that like wt Rad53, a kinase-dead mutant Rad53 is largely able to repress DDK binding to MCM, but the kinase-dead mutant Rad53 is unable to displace DDK from pre-formed DH. The authors suggest that their result would be explained if Rad53 dimerization is the essential requirement to inhibit DDK binding to Mcm2-7, mediated by Rad53 trans-autophosphorylation. While an interesting suggestion, they do not really test this idea. It is therefore suggested that the authors modulate the conclusions and model arising from Figure 6, to avoid the false impression that the necessity for Rad53 trans-auto-activation and dimerization has been demonstrated. One possibility would be to remove the data on the kinase-dead mutant altogether. Alternatively, the authors should considerably tone down their interpretation and discussion concerning the phenotype of the kinase-dead mutant (including removing the comment about autophosphorylation from the Abstract, and the illustration in Figure 7 that Rad53 phosphorylation and dimerization are important, since this has not been demonstrated).

---

## [Author Response]

Revisions:1) The level of DDK required for maximal binding to the Mcm2-7 DH and the phosphorylation of Mcm4 and Mcm6 is 75nM under the conditions employed, but Figure 5—figure supplement 1 shows that 300nM of DDK does not saturate the amount of DNA synthesis. Perhaps the authors could comment on this difference. Of course, DDK might perform multiple functions in the initiation of DNA replication.

We assume the reviewer was referring to the MCM DH-DDK binding data in Figure 4A, which demonstrate saturation of DDK binding to wildtype MCM DHs at 75 nM DDK. Please note that the experiment in Figure 5—figure supplement 1B has been carried out with the Mcm2-D127 truncation in order to demonstrate that increased DDK concentrations can partially suppress the replication defect of Mcm2-D127, consistent with the notion that the Mcm2 NTE is an affinity determinant for DDK. This experiment, therefore, corresponds to the MCM DH-DDK binding experiment in Figure 4B, lanes 5-8, which also does not reach saturation of DDK binding to MCM DHs even at 300 nM DDK.

Nonetheless, we do indeed observe that DNA synthesis does also not saturate at 75 nM DDK in replication assays with Mcm2-wt (data not shown, but available upon request). We believe this is due to the fact that in the replication assay DDK is added to a complex mixture of MCM DHs and free Cdt1·MCM complexes present at the MCM loading step of the replication reaction, whereas free Cdt1·Mcm2-7 complexes are removed by washes of the DNA beads in the MCM DH-DDK binding experiments of Figure 4 prior to addition of DDK. It is, therefore, likely that the excess of free Cdt1·MCM competes with DNA-bound MCM DHs for DDK in the replication assay.

2) In Figure 1, does the NTE truncation affect MCM loading (as well as DH stability)? This issue is actually addressed in Figure 5B, but the issue would usefully be covered at the start, either by including data in this figure or by making reference in the text to where the experiment will be presented.

We have moved panels A and B from the original Figure 5 to Figure 1 as new panels D+E and have made according changes to the main text.

3) Figure 3: Why does the efficiency of replication appear increased when TEV protease is added to MCM2-wt (Figure 3B)? Also, in Figure 3B, it would be useful to show a “-TEV protease” negative control within the Mcm2-TEV set of reactions.

The “-TEV protease” replication control experiment for Mcm2-TEV is shown in the original Figure 2—figure supplement 2.

4) Figure 4: Figure 4C: What concentration of DDK was used (to allow comparison with parts A and B)?

DDK was used at 150 nM. While previously noted in the Materials and methods section, we now indicate this also in the figure legend to Figure 4C for clarity.

Figure 4E: A band shift suggests that Dbf4 seems to be phosphorylated/modified when ATP is added. Could the authors explain, and discuss possible reasons and interpretation?

DDK is known to undergo autophosphorylation. This has been shown for both yeast and human DDK. We now provide appropriate references in the main text.

Could the authors explain in the text for what reason ATP/an ATP analogue might be needed to promote full levels of DDK association?

We can only speculate about the reasons, but now consider in the Discussion the possibility that by analogy to other protein kinases ATP binding either induces a conformational change or stabilizes a conformation in Cdc7 that is more conducive to substrate engagement.

5) Figure 5: Figure 5C: The text states that there is no evidence of asymmetric or uni-directional replication coming from the activation of a single hexamer, but the reasons given for this interpretation are difficult to understand. This conclusion needs to be better explained.

An expanded explanation is now included in the main text.

6) The weakest aspect of the paper revolves around the model that DDK bound Mcm2 is required for both of the two Mcm2-7 complexes for efficient initiation to occur. Although the authors propose a model that the 1:1 complex is inactive, the data is not sufficiently different from what would be expected if this complex were just slightly less active (e.g. 2-fold). More importantly, the model of that DDK bound to one kinase phosphorylates the tails of the other kinase has no support in the paper and should be removed or additional experiments added. If this trans model is right, then there are clear experiments to demonstrate this is the case that are well within the authors' abilities. They should be added to support the model or the model should be left out of the paper so that if or when they demonstrate this interesting idea it can be a clear new result. This is not an essential add but if such experiments are not added then the model should be removed. The authors would need more replicates of Figure 6 C, D, and E to be a convincing quantitative argument. Even then, there are clearly other explanations for the data.

We agree with the reviewer that additional evidence is needed to address the stoichiometry of the DDK-MCM DH complex and to resolve the issue of *cis*- or *trans*-phosphorylation of Mcm4/6 by DDK. In response to the reviewer’s comments we have, therefore, removed the interpretation of the data of Figure 5 with regard to DDK stoichiometry and refocused the description of the data in Figure 5 on the lack of uni-directional origin firing in the presence of both Mcm2-WT- and Mcm2D127. In addition, we now point out in the Discussion that it will be important in future studies to determine the stoichiometry of DDK in the DDK-MCM DH complex. At this point we do not make any assumptions in the paper regarding the *cis*-/*trans*-phosphorylation mechanism of Mcm4/6 by DDK.

7) Figure 6: It is shown that like wt Rad53, a kinase-dead mutant Rad53 is largely able to repress DDK binding to MCM, but the kinase-dead mutant Rad53 is unable to displace DDK from pre-formed DH. The authors suggest that their result would be explained if Rad53 dimerization is the essential requirement to inhibit DDK binding to Mcm2-7, mediated by Rad53 trans-autophosphorylation. While an interesting suggestion, they do not really test this idea. It is therefore suggested that the authors modulate the conclusions and model arising from Figure 6, to avoid the false impression that the necessity for Rad53 trans-auto-activation and dimerization has been demonstrated. One possibility would be to remove the data on the kinase-dead mutant altogether. Alternatively, the authors should considerably tone down their interpretation and discussion concerning the phenotype of the kinase-dead mutant (including removing the comment about autophosphorylation from the Abstract, and the illustration in Figure 7 that Rad53 phosphorylation and dimerization are important, since this has not been demonstrated).

We agree with the reviewer that the role of Rad53 autophosphorylation and dimerization needs to be affirmed by additional data. This is complicated by the fact that activation/autophosphorylation depends on Rad53 dimerization. In light of the reviewer’s comments we have removed the statement on the roles of Rad53 autophosphorylation and dimerization from both the Abstract and the conclusions in the main text. Instead, we raise the possibility in the Discussion that the Rad53 oligomeric state or its phosphorylation state, or both, may control the ability of Rad53 to sequester DDK.

Moreover, we now present additional data demonstrating that Rad53-wt, but not Rad53-kd, can form a stable complex with DDK, which supports our model for the steric inhibition of DDK by Rad53. In addition, while our previous experiments have focused on the effect of Rad53 on DDK binding to MCM DHs, we now include data that extend the effect of Rad53 on DNA synthesis. We consider our observation of a steric mechanism for Rad53 function a major novel finding and consider the data involving Rad53-kd important to support this observation.